# Grokking Beyond Neural Networks: An Empirical Exploration with Model Complexity

**Jack Miller** *jack.miller@anu.edu.au*
*ANU College of Engineering, Computing and Cybernetics*

**Charles O'Neill** *charles.oneill@anu.edu.au*
*ANU College of Engineering, Computing and Cybernetics*

**Thang Bui** *thang.bui@anu.edu.au*
*ANU College of Engineering, Computing and Cybernetics*

**Reviewed on OpenReview:** *https://openreview.net/forum?id=ux9BrxPCl8*

## Abstract

In some settings neural networks exhibit a phenomenon known as *grokking*, where they achieve perfect or near-perfect accuracy on the validation set long after the same performance has been achieved on the training set. In this paper, we discover that grokking is not limited to neural networks but occurs in other settings such as Gaussian process (GP) classification, GP regression, linear regression and Bayesian neural networks. We also uncover a mechanism by which to induce grokking on algorithmic datasets via the addition of dimensions containing spurious information. The presence of the phenomenon in non-neural architectures shows that grokking is not restricted to settings considered in current theoretical and empirical studies. Instead, grokking may be possible in any model where solution search is guided by complexity and error.

## 1 Introduction

In this paper, we conduct an empirical exploration of grokking, uncovering new aspects of the phenomenon not explained by current theory. We begin by describing grokking and summarise its existing explanations. Afterwards, we present our empirical observations – most notably, the existence of grokking outside of neural networks. Finally, we suggest a mechanism for grokking that is broadly consistent with our observations. While we do not claim that this mechanism is necessary in all cases of grokking as it relies upon a complexity penalty, we do provide evidence that it may be necessary in some of our learning settings.

### 1.1 Generalisation

Grokking is defined with respect to model generalisation – the capacity to make good predictions in novel scenarios. To express this notion formally, we restrict our definition of generalisation to supervised learning. In this paradigm, we are given a set of training examples $X \subset \mathcal{X}$ and associated targets $Y \subset \mathcal{Y}$. We then attempt to find a function $f_\theta : \mathcal{X} \to \mathcal{Y}$ so as to minimise an objective function $\mathcal{L} : (X, Y, f_\theta) \to \mathbb{R}$.[1] Having found $f_\theta$, we may then look at the function's performance under a possibly new objective function $\mathcal{G}$ (typically $\mathcal{L}$ without dependence on $\theta$) on an unseen set of examples and labels, $X'$ and $Y'$. If $\mathcal{G}(X', Y')$ is small, we say that the model has generalised well and if it is large, it has not generalised well.

---

[1]A popular choice for $f_\theta$ is a neural network (Goodfellow et al., 2016) with $\theta$ representing its weights and biases.

## 1.2 Model Selection and Complexity

Consider $\mathscr{F}$, a set of models we wish to use for prediction. We have described a process by which to assess the generalisation performance of $f_\theta \in \mathscr{F}$ given $X'$ and $Y'$. Unfortunately, these hidden examples and targets are not available during training. As such, we may want to measure relevant properties about members of $\mathscr{F}$ that could be indicative of their capacity for generalisation. One obvious property is the value of $\mathcal{G}(X, Y)$ (or some related function) which we call the *data fit*. Another property is *complexity*. If we can find some means to measure the complexity, then traditional thinking would recommend we follow the principle of parsimony. "[The principle of] parsimony is the concept that a model should be as simple as possible with respect to the induced variables, model structure, and number of parameters" (Burnham & Anderson, 2004). That is, if we have two models with similar data fit, we should choose the simplest of the two.

If we wish to follow both the principle of parsimony and minimise the data fit, the loss function $\mathcal{L}$ we use for choosing a model is given by:

$$\mathcal{L} = \text{error} + \text{complexity}. \tag{1}$$

While Equation 1 may seem simple, it is often difficult to characterise the complexity term[2]. Not only are there multiple definitions for model complexity across different model categories, definitions also change among the same category. When using a decision tree, we might measure complexity by tree depth and the number of leaf nodes (Hu et al., 2021). Alternatively, for deep neural networks, we could count the number and magnitude of parameters or use a more advanced measure such as the linear mapping number (LMN) (Liu et al., 2023b). Fortunately, in this wide spectrum of complexity measures, there are some unifying formalisms we can apply. One such formalism was developed by Kolmogorov (Yueksel et al., 2019). In this formalism, we measure the complexity as the length of the minimal program required to generate a given model. Unfortunately, the difficulty of computing this measure makes it impractical. A more pragmatic alternative is the model description length. This defines the complexity of a model as the minimal message length required to communicate its parameters between two parties (Hinton & van Camp, 1993). We discuss Kolmogorov complexity in Appendix B and the model description length in Appendix C. We also provide a note on measuring complexity in GPs which can be found in Appendix D.

## 1.3 The Grokking Phenomenon

Grokking was recently discovered by Power et al. (2022) and has garnered attention from the machine learning community. It is a phenomenon in which the performance of a model $f_\theta$ on the training set reaches a low error at epoch $E_1$, then following further optimisation, the model reaches a similarly low error on the validation dataset at epoch $E_2$. Importantly the value $\Delta_k = |E_2 - E_1|$ must be non-trivial. In many settings, the value of $\Delta_k$ is much greater than $E_1$. Additionally, the change from poor performance on the validation set to good performance can be quite sudden.[3] A prototypical illustration of grokking is provided in Figure 13 (Appendix J.2).

To the best of our knowledge, all notable existing empirical literature on the grokking phenomenon is summarised in Table 1 (Appendix A). The literature has focused primarily on neural network architectures and algorithmic datasets[4]. We use a variety of similar algorithmic datasets outlined in Appendix E.

No paper has yet demonstrated the existence of the phenomenon using a GP. In addition, several theories have been presented to explain grokking. They can be categorised into three main classes based on the mechanism they use to analyse the phenomenon. Loss-based theories such as Liu et al. (2023a) appeal to the loss landscape of the training and test sets under different measures of complexity and data fit. Representation-based theories such as Davies et al. (2023), Barak et al. (2022), Nanda et al. (2023) and Varma et al. (2023) claim that grokking occurs as a result of feature learning (or circuit formation) and associated training dynamics. Finally, there are a set of theories which use the neural tangent kernel (NTK)

---

[2]There exist standard choices for $\mathscr{G}(X, Y)$ such as the cross entropy in classification or the $L_2$ norm in regression

[3]Whether this sharp transition is required to fulfil the definition of grokking is somewhat unclear. In this paper, we tend to accept both sharp and soft transitions.

[4]We define an algorithmic dataset as one in which labels are produced via a predefined algorithmic process such as a mathematical operation between two integers.

(Jacot et al., 2018) to explain grokking. Namely, Kumar et al. (2023) and Lyu et al. (2023). Importantly, no existing theory could explain grokking if it were found in GPs.

**Loss based theory.** Liu et al. (2023a) assume that there is a spherical shell (Goldilocks zone) in the weight space where generalisation is better than outside the shell. They claim that, in a typical case of grokking, a model will have large weights and quickly reach an over-fitting solution outside of the Goldilocks zone. Then regularisation will slowly move weights towards the Goldilocks zone. That is, grokking occurs due to the mismatch in time between the discovery of the overfitting solution and the general solution. While some empirical evidence is presented in Liu et al. (2023a) for their theory, and the mechanism itself seems plausible, the requirement of a spherical Goldilocks zone seems too stringent. It may be the case that a more complicated weight-space geometry is at play in the case of grokking.

Liu et al. (2023b) also recently explored some cases of grokking using the LMN metric. They find that during periods they identify with generalisation, the LMN decreases. They claim that this decrease in LMN is responsible for grokking.

**Representation or circuit based theory.** Representation or circuit based theories require the emergence of certain general structures within neural architectures. These general structures become dominant in the network well after other less general ones are sufficient for low training loss. For example, Davies et al. (2023) claim that grokking occurs when, "slow patterns generalize well and are ultimately favoured by the training regime, but are preceded by faster patterns which generalise poorly." Similarly, it has been shown that stochastic gradient descent (SGD) slowly amplifies a sparse solution to algorithmic problems which is hidden to loss and error metrics (Barak et al., 2022). This is mirrored somewhat in the work of Liu et al. (2022) which looks to explain grokking via a slow increase in representation quality[5]. Nanda et al. (2023) claim in the setting of an algorithmic dataset and transformer architecture, training dynamics can be split into three phases based on the network's representations: memorisation, circuit formation and cleanup. Additionally, the structured mechanisms (circuits) encoded in the weights are gradually amplified with later removal of memorising components. Varma et al. (2023) are in general agreement with Nanda et al. (2023). Representation (or circuit) theories seem the most popular, based on the number of research papers published which use these ideas. They also seem to have a decent empirical backing. For example, Nanda et al. (2023) explicitly discover circuits in a learning setting where grokking occurs.

**NTK based theory**. Recently, Kumar et al. (2023) and Lyu et al. (2023) used the neural tangent kernel to explain the grokking phenomenon. Specifically, Kumar et al. (2023) found that a form of grokking still occurs without an explicit complexity penalty. They argue that in this case, grokking is caused by the transition between lazy and feature-rich training regimes. This is distinct from previous theories of grokking in neural networks, many of which require weight decay to remove "memorising" solutions. Lyu et al. (2023) worked concurrently on grokking with small values of weight decay, demonstrating that under idealised conditions, grokking could be provably induced through a sharp transition from an early kernel regime to a rich regime.

**Grokking in Linear Estimators**. Concurrently to the writing of this paper, Levi et al. (2023) studied grokking in a linear student-teacher setting. In that work they proved that a significant gap between the training and validation accuracy can arise when the two networks are given Gaussian inputs. This shows that grokking can occur in linear settings without the need for a complex model that might transition from "memorisation" to "understanding."

**Our contributions.** Unlike previous work, in this paper, we choose another axis by which to explore grokking. We restrict ourselves to cases where the loss function can be decomposed into the form of Equation 1 but expand the set of models we study. In particular, we contribute to the existing grokking literature in the following ways:

- We demonstrate that grokking occurs when tuning the hyperparameters of GPs and in the case of linear models. This necessitates a new theory of grokking outside of neural networks.

- We create a new data augmentation technique which we call concealment that can increase the grokking gap via control of additional spurious dimensions added to input examples.

---

[5]See Figure 1 of this paper for a high quality visualisation of what is meant by a general representation.

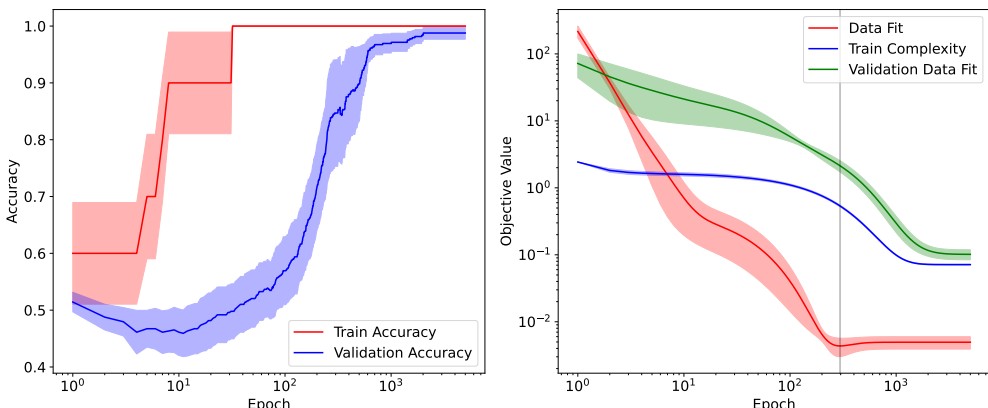

Figure 1: Accuracy, data fit and complexity on zero-one slope classification task with a linear model. Note that the shaded region corresponds to the standard error of five training runs. Further, the grey line marks the point of minimum data fit.

- We suggest a mechanism for grokking in cases where solution search is guided by complexity and error. This mechanism is model-agnostic and can explain grokking outside of neural networks.

## 2 Experiments

In the following section, we present various empirical observations we have made regarding the grokking phenomenon. In Section 2.1, we demonstrate that grokking can occur with GP classification and linear regression. This proves its existence in non-neural architectures, identifying a need for a more general theory of the phenomenon. Further, in Section 2.2, we show that there is a way of inducing grokking via data augmentation. Finally, in Section 2.3, we examine directly the weight-space trajectories of models which grok during training, incidentally demonstrating the phenomenon in GP regression. Due to their number, the datasets used for these experiments are not described in the main text but rather in Appendix E.

### 2.1 Grokking in GP Classification and Linear Regression

In the following experiments, we show that grokking occurs with GP classification and linear regression. In the case of linear regression, a very specific set of circumstances were required to induce grokking. However, for the GP models tested using typical initialisation strategies, we were able to observe behaviours which are consistent with our definition of grokking.

#### 2.1.1 Zero-One Classification on a Slope with Linear Regression

As previously mentioned, to demonstrate the existence of grokking with linear regression, a very specific learning setting was required. We employed Dataset 9 with three additional spurious dimensions and two training points. Given a particular example $x_0$ from the dataset, the spurious dimensions were added as follows to produce the new example $x'$:

$$x' = \begin{bmatrix} x_0 & x_0^2 & x_0^3 & \sin(100x_0) \end{bmatrix}^T \tag{2}$$

The model was trained as though the problem were a regression task with outputs later transformed into binary categories based on the sign of the predictions.[6] To find the model weights, we used SGD over a standard loss function with a mean squared error (MSE) data fit component and a scaled weight-decay

---

[6]If $f_\theta(x) < 0$ then the classification of a given point was negative and if $f_\theta(x) > 0$, then the class was positive.

complexity term[7]:

$$L = \frac{\text{MSE}(y, \hat{y})}{\epsilon_0} + \sum_{i=0}^{d} \frac{(w^i - \mu_0^i)^2}{\sigma_0^i}. \tag{3}$$

In Equation 3, the term $w^i$ represents the individual weights of the linear regression model corresponding to each feature dimension. Further, $i$ indexes the dimensionality of the variables $w$, $\mu_0$ and $\sigma_0$, $\epsilon_0$ is the noise variance, $\mu_0$ is the prior mean, and $\sigma_0$ is the prior variance. For our experiment, $\mu_i$ was taken to be 0 and $\sigma_i$ to be 0.5 for all $i$. Regarding initialisation, $w$ was heavily weighted against the first dimension of the input examples[8]. This unusual alteration to the initial weights was required for a clear demonstration of grokking with linear regression.

The accuracy, complexity and data fit of the linear model under five random seeds governing dataset generation is shown in Figure 1. Clearly, in the region between epochs $2 \cdot 10^1$ and $10^3$, validation accuracy was significantly worse than training accuracy and then, in the region $2 \cdot 10^3$, the validation accuracy was very similar to that of the training accuracy. This satisfies our definition of grokking, although the validation accuracy did not always reach 100% in every case. Provided in Appendix H.1 is the accuracy and loss of a training run with a specific seed that reaches 100% accuracy.

We also completed a series of further experiments concerning model initialisation, weight evolution and the necessity of weight decay which we present in Appendix K. We find that the resulting trends are consistent with the grokking mechanism we suggest in Section 3 and so are the evolution of the weights. Further, we show that without weight decay we do not see grokking. This demonstrates that some form of regularisation is required, providing evidence that the grokking mechanism we suggest might be both sufficient and necessary in this setting.

### 2.1.2 Zero-One Classification with a Gaussian Process

In our second learning scenario, we applied GP classification to Dataset 8 with a radial basis function (RBF) kernel:

$$k(x_1, x_2) = \alpha \exp\left(-\frac{1}{2}(x_1 - x_2)^T \Theta^{-2} (x_1 - x_2)\right). \tag{4}$$

Here, $\Theta$ is called the lengthscale parameter and $\alpha$ is the kernel amplitude. Both $\Theta$ and $\alpha$ were found by minimising the approximate negative marginal log likelihood associated with a Bernoulli likelihood function via the Adam optimiser acting over the variational evidence lower bound (Hensman et al., 2015; Gardner et al., 2018).

The result of training the model using five random seeds for dataset generation and model initialisation can be seen in Figure 2. The final validation accuracy is not 100% like the cases we will see in the proceeding sections. However, it is sufficiently high to say that the model has grokked. In Appendix H.2, we also provide the loss curves during training. As we later discuss in Section 4, there are some subtleties associated with interpreting the role of the complexity term.

In our third learning scenario, we also looked at GP classification. However, this time on a more complex algorithmic dataset – a modified version of Dataset 1 (with $k = 3$) where additional spurious dimensions are added and populated using values drawn from a normal distribution. In particular, the number of additional dimensions is $n = 37$ making the total input dimensionality $d = 40$. We use the same training setup as in Section 2.1.2. The results (again with five seeds) can be seen in Figure 3 with a complexity plot in Appendix H.3 and discussion of the limitations of this complexity measure in Section 4.

For both GP learning scenarios we also completed experiments without the complexity term arising under the variational approximation. The results of these experiments, namely a lack of grokking, can be seen in Appendix L. This demonstrates that some form of regularisation is needed in this scenario and provides further evidence for the possible necessity of the grokking mechanism we propose in Section 3.

---

[7]As demonstrated by Hinton & van Camp (1993), this loss function has an equivalence with the MDL principle when we fix the standard deviations of prior and posterior Gaussian distributions.

[8]$w^0 = \begin{bmatrix} 5 \cdot 10^{-4} & 9 \cdot 10^{-1} & 9 \cdot 10^{-1} & 9 \cdot 10^{-1} \end{bmatrix}^T$.

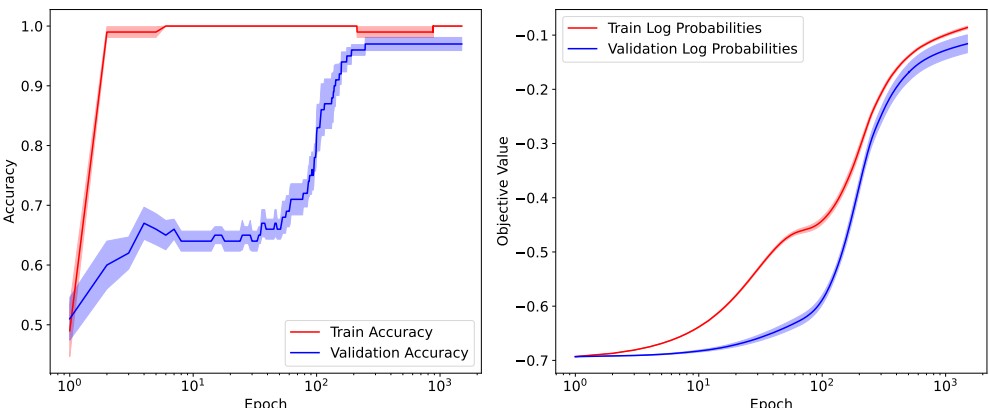

Figure 2: Accuracy and log likelihoods on zero-one classification task with a RBF Gaussian process. Note that the shaded region corresponds to the standard error of five training runs.

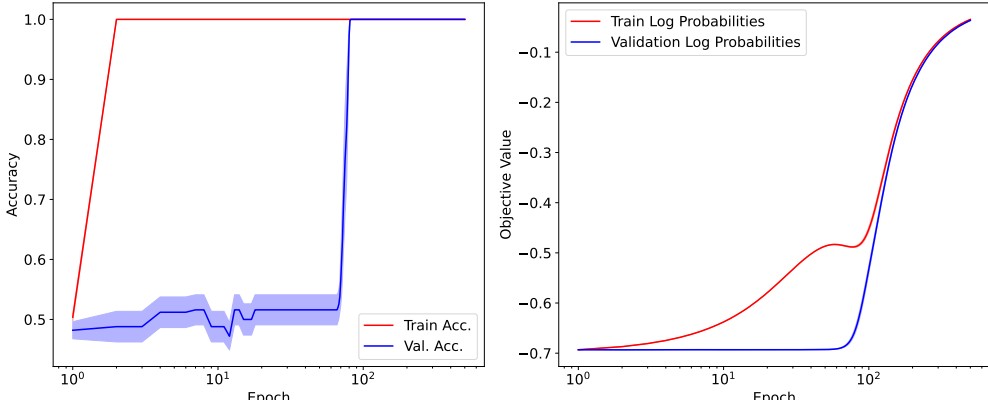

Figure 3: Accuracy and log likelihoods on hidden parity prediction task with RBF Gaussian process. Note that the shaded region corresponds to the standard error of five training runs. *Acc.* is *Accuracy* and *Val.* is *Validation.*

## 2.2 Inducing Grokking via Concealment

In this section, we investigate how one might augment a dataset to induce grokking. In particular, we develop a strategy which induces grokking on a range of algorithmic datasets. This work was inspired by Merrill et al. (2023) and Barak et al. (2022) where the true task is "hidden" in a higher dimensional space. This requires models to "learn" to ignore the additional dimensions of the input space. For an illustration of *learning to ignore* see Figure 12 (Appendix J.1).

Our strategy is to extend this "concealment" idea to other algorithmic datasets. Consider $(x, y)$, an example and target pair in supervised learning. Under concealment, one augments the example $x$ by adding random bits, labeled $v_0$ to $v_l$, where the values 0 and 1 have equal probability. The new concealed example $x'$ is:

$$x' = \begin{bmatrix} x_0 & x_1 & \cdots & x_d & v_0 & \cdots & v_l \end{bmatrix}^T.$$

To determine the generality of this strategy in the algorithmic setting, we applied it to 6 different datasets (2-7). These datasets were chosen as they share a regular form[9] and seem to cover a fairly diverse variety of algorithmic operations. In each case, we used the prime 7, and varied the additional dimensionality $k$. For the model, we used a simple neural network analogous to that of Merrill et al. (2023). This neural network

---

[9]They are all governed by the same prime $p$ and take two input numbers.

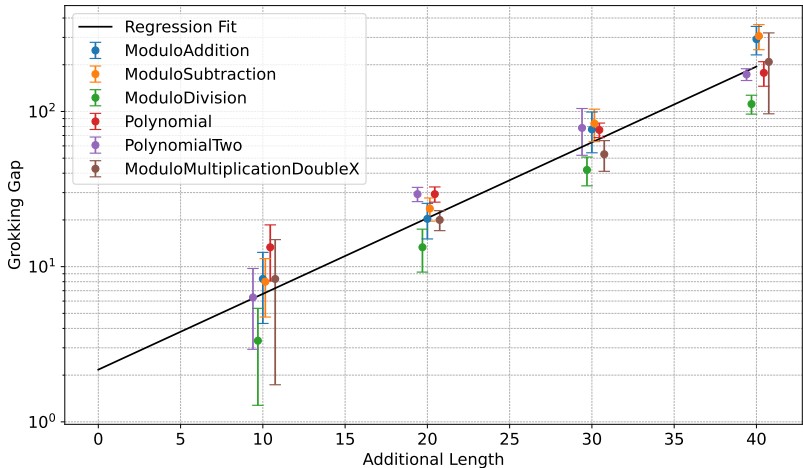

Figure 4: Relationship between grokking gap and number of additional dimensions using the grokking via concealment strategy. Note that $x$-values are artificially perturbed to allow for easier visibility of error bars. In reality they are either 10, 20, 30 or 40. Also, the data of zero additional length is removed (although still influences the regression fit). See Appendix I.2 for the plot without these changes.

consisted of 1 hidden layer of size 1000 and was optimised using SGD with cross-entropy loss. The weight decay was set to $10^{-2}$ and the learning rate to $10^{-1}$. Loss plots for all experiments are shown in Appendix N.

To discover the relationship between concealment and grokking, we measured the "grokking gap" $\Delta_k$. In particular, we considered how an increase in the number of spurious dimensions relates to this gap. The algorithm used to run the experiment is detailed in Algorithm 1 (Appendix I.1). The result of running this algorithm can be seen in Figure 4. In addition to visual inspection of the data, a regression analysis was completed to determine whether the relationship between grokking gap and additional dimensionality might be exponential. The details of the regression are provided in Appendix F and its result is denoted as *Regression Fit* in the figure. The Pearson correlation coefficient (Pearson, 1895; SciPy developers, 2023) was also calculated in log space for all points available and for each dataset individually. Further, we completed a test of the null hypothesis that the distributions underlying the samples are uncorrelated and normally distributed. The Pearson correlation $r$ and $p$-values are presented in Table 2 (Appendix F.2) The Pearson correlation coefficients are high in aggregate and individually, indicating a positive linear trend in log space. Further, $p$ values in both the aggregate and individual cases are well below the usual threshold of $\alpha = 0.05$.

In Appendix M, one can also find an analysis of this data augmentation technique with regards to the grokking mechanism we suggest in Section 3. This analysis involved scaling the magnitude of weight matrices to determine how this would alter trends seen in Figure 4.

## 2.3 Parameter Space Trajectories of Grokking

Our last set of experiments was designed to interrogate the parameter space of models which grok. We completed this kind of interrogation in two different settings. The first was GP regression on Dataset 10 and the second was BNN classification on a concealed version of Dataset 1. Since the GP only had two hyperparameters governing the kernel, we could see directly the contribution of complexity and data fit terms. Alternatively, for the BNN we aggregated data regarding training trajectories across several initialisations to investigate the possible dynamics between complexity and data fit.

### 2.3.1 GP Grokking on Sinusoidal Example

In this experiment, we applied a GP (with the same kernel as in Section 2.1.2) to regression of a sine wave. To find the optimal parameters for the kernel, a Gaussian likelihood function was employed with

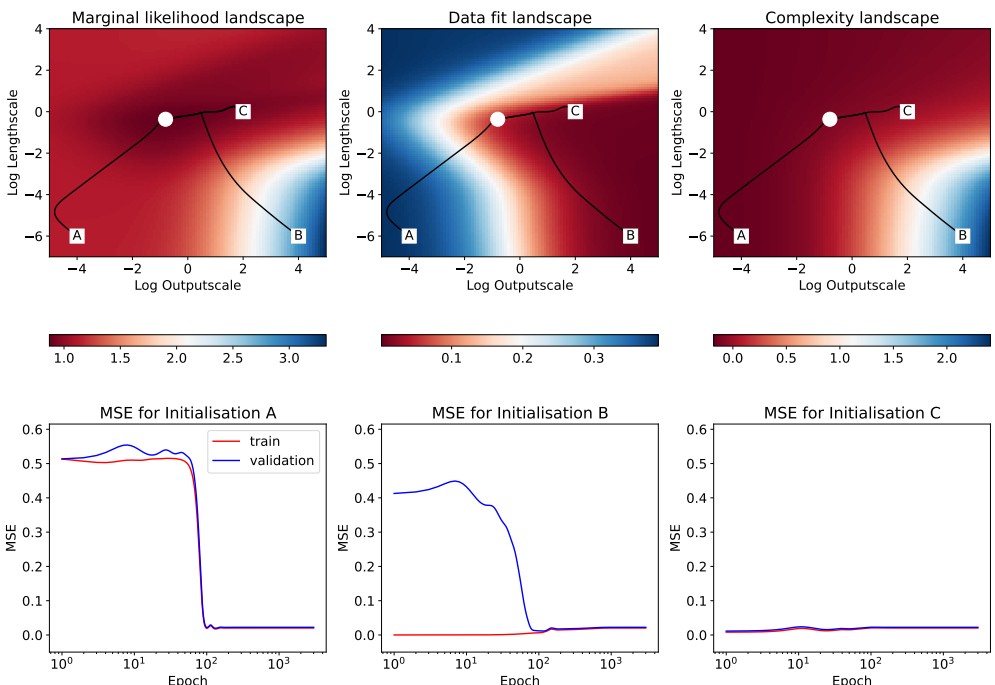

Figure 5: Trajectories through parameter landscape for GP regression. Initialisation points A-C refer to those mentioned in Section 2.3.1.

exact computation of the marginal log likelihood. In this optimisation scenario, the complexity term is as described in Appendix D.

To see how grokking might be related to the complexity and data fit landscapes, we altered hyperparameter initialisations. We considered three different initialisation types. In case A, we started regression in a region of high error and low complexity (HELC) where a region of low error and high complexity (LEHC) was relatively inaccessible when compared a region of low error and low complexity (LELC). For case B, we initialised the model in a region of LEHC where LELC solutions were less accessible. Finally, in case C, we initialised the model in a region of LELC. As evident in Figure 5, we only saw grokking for case B. It is interesting that, in this GP regression case, we did not see a clear example of the spherical geometry mentioned in the Goldilocks zone theory of Liu et al. (2023a). Instead, a more complicated loss surface is present which results in grokking.

### 2.3.2   Trajectories of a BNN with Parity Prediction

We also examined the weight-space trajectories of a BNN ($f_\theta$). Our learning scenario involved Dataset 1 with the concealment strategy presented in Section 2.2. Specifically, we used an additional dimensionality of 27 and a parity length of 3. To train the model, we employed SGD with the following variational objective:

$$\mathcal{L}(\phi) = \mathbb{E}_{Q_\phi(\theta)}[\text{CrossEntropyLoss}(f_\theta(X), Y)] + D_{KL}(Q_\phi(\theta)||P(\theta)). \tag{5}$$

In Equation 5, $P(\theta)$ is a standard Gaussian prior on the weights and $Q_\phi(\theta)$ is the variational approximation. The complexity penalty in this case is exactly the model description length discussed in Section 1.2 with the overall loss function clearly a subset of Equation 1.

To explore the weight-space trajectories of the BNN we altered the network's initialisation by changing the standard deviation of the normal distribution used to seed the variational mean of the weights. This resulted in network initialisations with differing initial complexity and error. We then trained the network based on these initialisations using three random seeds, recording values of complexity, error and accuracy. The outcomes of this process are in Figure 6. Notably, initialisations which resulted in an increased grokking

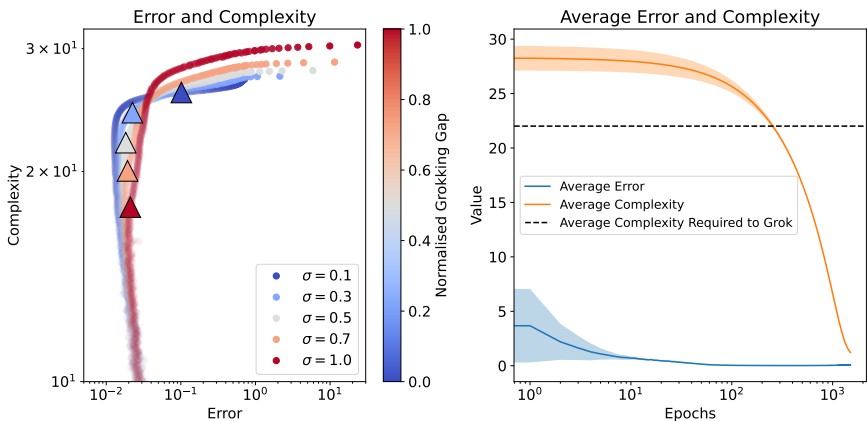

Figure 6: This figure illustrates grokking in Bayesian Neural Networks (BNNs) using different standard deviation values ($\sigma$) for variational mean initialisations. The left plot shows the error-complexity landscape during training. Points become less transparent as training progresses, indicating epoch advancement. Triangles mark the grokking point for each $\sigma$, determined when validation accuracy first exceeds 95% after training accuracy reaches this level. Three trials per $\sigma$ contribute to these points. The right plot displays the average error and complexity across epochs, highlighting how complexity evolves during the training and grokking phases. *Normalised Grokking Gap* refers to the epoch difference between achieving 95% training and validation accuracy, scaled between zero and one.

gap correlate with increased optimisation time in regions of LEHC. Further, there seems to be a trend across epochs with error and complexity. At first, there is a significant decrease in error followed by a decrease in complexity and it is in this region, where complexity is decreasing most, that grokking occurs.

## 3    Grokking and Complexity

So far we have explored grokking with reference to different complexity measures across a range of models. There seems to be no theory of grokking in the literature which can explain the new empirical evidence we present. Motivated by this, we suggest a possible mechanism which may induce grokking in cases where solution search is guided by both complexity. We believe this condition to be compatible with our new results, previous empirical observations and with many previous theories of the grokking phenomenon.

To build our hypothesised grokking mechanism, we first make Assumption 1. We believe that Assumption 1 is justified for the most common setting in which grokking occurs. Namely, algorithmic datasets. It is also likely true for a wide range of other scenarios.

**Assumption 1.** *For the task of interest, the principle of parsimony holds. That is, solutions with minimal possible complexity will generalise better.*

We then suggest Mechanism 1 which we refer to as the *complexity route to grokking.*

**Mechanism 1.** *If the low error, high complexity (LEHC) weight space is readily accessible from typical initialisation but the low error, low complexity (LELC) weight space is not, models will quickly find a low error solution which does not generalise. Given a path between LEHC and LELC regions which has non-increasing loss, solutions in regions of LEHC will slowly be guided toward regions of LELC due to regularisation. This causes an eventual decrease in validation error, which we see as grokking.*

We do not claim that this is an exclusive route to the grokking phenomenon. Kumar et al. (2023) and others have demonstrated that grokking does not require an explicit regularisation scheme in the neural network cases. However, we believe it could be a sufficient mechanism there. In addition, it may be the dominant mechanism in the non-neural network cases we present. Evidence for this is provided in Appendices K and L where we demonstrate the need for regularisation in our linear regression and GP grokking cases.

### 3.1 Explanation of Previous Empirical Results

We now discuss the congruence between our hypothesised mechanism and existing empirical observations. In Appendix G we draw parallels between our work and existing theory.

Learning with algorithmic datasets benefits from the principle of parsimony as a small encoding is required for the solution. In addition, when learning on these datasets, there appear to be many other more complex solutions which do not generalise but attain low training error. For example, with a neural network containing one hidden layer completing a parity prediction problem, there is competition between dense subnetworks which are used to achieve high accuracy on the training set (LEHC) and sparse subnetworks (LELC) which have better generalisation performance (Merrill et al., 2023). In this case, the reduced accessibility of LELC regions compared to LEHC regions seems to cause the grokking phenomenon. This general story is supported by further empirical analysis completed by Liu et al. (2022) and Nanda et al. (2023). Liu et al. (2022) found that a less accessible, but more general representation, emerges over time within the neural network they studied and that after this representation's emergence, grokking occurs. If the principle of parsimony holds, this general representation should be simpler under an appropriate complexity measure. Consequently, the shift could be explained by Mechanism 1. Nanda et al. (2023) discovered that a set of trigonometric identities were employed by a transformer to encode an algorithm for solving modular arithmetic. Additionally, this trigonometric solution was gradually amplified over time with the later removal of high complexity "memorising" structures. Indeed, it is stated in that paper that circuit formation likely happens due to weight decay. This fits under Mechanism 1 – the model is moving from an accessible LEHC region where memorising solutions exist to a LELC region via regularisation.

The work by Liu et al. (2023a) showed the existence of grokking on non-algorithmic datasets via alteration of the initialisation and dataset size. From Mechanism 1, we can see why these factors would alter the existence of grokking. Changing the initialisation alters the relative accessibility of LEHC and LELC regions and reducing the dataset size may lessen constraints on LEHC regions which otherwise do not exist.

The study by Levi et al. (2023) on grokking in linear models complements our hypothesis on grokking's nature, particularly in the linear regression context. Their focus on the divergence between training and generalisation loss in linear teacher-student models, influenced by input dimensionality and weight decay, parallels our findings in inducing grokking via dataset manipulation. Our approach, especially with Dataset 9, where we add spurious dimensions, aligns with their emphasis on input structure's role. This similarity bolsters our claim that grokking emerges due to the initial inaccessibility of LELC solutions. Furthermore, our exploration in GP models through data concealment, adding uninformative dimensions, demonstrates that these relationships extend into more complex scenarios beyond linear estimators.

### 3.2 Explanation of New Empirical Evidence

Having been proposed to explain the empirical observation we have uncovered in this paper, Mechanism 1 should be congruent with these new findings – the first of which is the existence of grokking in non-neural models. Indeed, one corollary of our theory (Corollary 1) is that grokking should be model agnostic. This is because the proposed mechanism only requires certain properties of error and complexity landscapes during optimisation. It is blind to the specific architecture over which optimisation occurs.

**Corollary 1.** *The phenomenon of grokking should be model agnostic. Namely, it could occur in any setting in which solution search is guided by complexity and error.*

Another finding from this paper is that of the concealment data augmentation strategy. We believe this can be explained via the lens of Mechanism 1 as follows. When dimensions are added with uninformative features, there exist LEHC solutions which use these features. However, the number of LELC solutions remains relatively low as the most general solution should have no dependence on the additional components. This leads to an increase in the relative accessibility of LEHC regions when compared to LELC regions which in turn leads to grokking.

## 4 Discussion

Despite some progress made toward understanding the grokking phenomenon in this paper, there are still some points to discuss. We start by assessing the limitations of the empirical evidence gathered. This is important for a balanced picture of the experimentation completed and its implications for our suggested grokking mechanism. Having examined these limitations, we can provide some recommendations regarding related future work in the field.

### 4.1 Limitations of Empirical Evidence

In Section 2.1, experimentation with linear regression may be criticised for the specificity of the learning setup required to demonstrate grokking and for the value of the final validation accuracy. We note that the first critique is not reasonable in the sense that we should be able to show grokking under "normal" circumstances since grokking does not appear under "normal circumstances." However, if one wanted to widen the scope of learning settings where Mechanism 1 applies, further experimentation is needed. It could be the case that, with only one setting, we saw results consistent with Mechanism 1 but under another learning scenario, our mechanism could become inconsistent. We do not consider the second critique to be significant. For our purposes, grokking need not have 100% accuracy as not all general solutions provide that. However, if this is desired, we provide a case where this occurs in Appendix H.1.

There are also reasonable critiques concerning the experimentation completed on GP classification. The most pressing might be concerns over the measurement of complexity as presented in Appendices H.2 and H.3. This is due to the way the model is optimised. Namely, via maximisation of the evidence lower bound:

$$\mathcal{L}_{\text{ELBO}}(\phi, \theta) = \sum_{i=1}^{N} \mathbb{E}_{q_\phi(f_i)}[\log p(y_i|f_i)] - \beta \text{KL}[q_\phi(f)||p_\theta(f)]. \tag{6}$$

Unfortunately, optimisation of this value leads to changes in both the variational approximation and the hyperparameters of the prior GP. This presents a problem when trying to use the results of GP classification to validate Mechanism 1. The hyperparameters control the complexity of the prior which then influences the measured complexity of the model via the KL divergence. Consequently, the complexity measurement at any two points in training are not necessarily comparable. To disentangle optimisation of the hyperparameters and the variational approximation, one could complete a set of ablation studies. For this, one would keep either the hyperparameters or the variational approximation constant and alter the other variable. By doing so, one would be able to validate more directly Mechanism 1 with GP classification. Additionally, one might need to alter the learning setting to retain grokking under a new approximation scheme such as Laplace's method. Further discussion and experimentation are provided in Appendix O.

Due to the simplicity of the model considered in Figure 5, it is hard to criticise experimentation completed there. However, the experimental design of the BNN lacks generality. Indeed, it is difficult to know if the subspace from which the BNN was initialised is indicative of general trends about the weight space. However, the values chosen were indicative of typical initialisation values. Thus, we can say that for "normal" cases that might be encountered by a practitioner, the BNN weight trajectories are representative.

### 4.2 Future Work

An interesting outcome of experimentation in this paper was the discovery of the concealment data augmentation strategy. As far as the authors are aware, this is the first data augmentation strategy found which consistently results in grokking. Additionally, its likely exponential trend with the degree of grokking is of great interest. Indeed, we know that the volume of a region in an $n$-dimensional space decreases exponentially with an increase in $n$. This fact and the exponential increase in grokking with additional dimensionality could be connected. Unfortunately, at this point in time, such a connection is only speculative. Thus, a more theoretical analysis might be warranted which seeks to examine this relationship.

Additional work should also be conducted to investigate the need for regularisation in grokking. In our work, we have suggested a mechanism for grokking which requires a complexity penalty. We believe this

mechanism to be broadly compatible with previous experimentation and theory produced for cases with such a penalty. However, the work completed by Levi et al. (2023) and Kumar et al. (2023) show that grokking need not require a complexity penalty in every case. Through the novel ablation studies provided, we have begun to draw a boundary around cases where grokking does seem to need a penalty like weight decay or KL divergence. Nonetheless, there is much still left to do in order to investigate the role of explicit regularisation in grokking.

### 4.3 The Real-World Applications of Grokking

We would also like to briefly touch upon the possible applications of our work on grokking. We believe there are several ways it may be useful. The most obvious case would be the discovery of grokking in datasets important to ML practitioners. In this scenario, practitioners could leverage our analysis to mitigate grokking by, for example, modifying their initialisation strategy. Indeed, our paper is the first to examine grokking in GPs, offering unique insights relevant to grokking in that model class. Another important consideration is the relationship between grokking and concealment. Given that some form of concealment could occur in real-world tasks, it merits analysis to determine whether any form of grokking is present and, if absent, to understand why.

## 5 Conclusion

We have presented novel empirical evidence for the existence of grokking in non-neural architectures and discovered a data augmentation technique which induces the phenomenon. Relying upon these observations and analysis of training trajectories in a GP and BNN, we suggested a mechanism for grokking in models where solution search is guided by complexity and error. Importantly, we argued that this theory is congruent with previous empirical evidence and many previous theories of grokking. In future, researchers could extend the ideas in this paper by undertaking a theoretical analysis of the concealment strategy discovered and by conducting further studies to assess the role of complexity penalties.

### Supplementary Material

All experiments can be found at this GitHub page. They have descriptive names and should reproduce the figures seen in this paper. For Figure 6, the relevant experiment is in the `feat/info-theory-description` branch.

### Broader Impact Statement

We have completed empirical experiments with the grokking phenomenon using synthetic data. It is difficult to identify any specific ethical concerns. Of course, broader ethical reservations, present in any basic machine learning research, might be relevant.

### Acknowledgements

We would also like to acknowledge Russell Tsuchida, Matthew Ashman, Rohin Shah, Yuan-Sen Ting, Oliver Balfour and Yashvir Grewal for their valuable input. In addition we would like to thank the Tuckwell scholarship program for funding the undergraduate studies of Jack Miller and Charles O'Neill.

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

# A    Summary of Existing Empirical Work on Grokking

| Research Paper | Architecture | Category | Dataset Description |
|---|---|---|---|
| Power et al. (2022) | Transformer | Algorithmic | Problems of the form $a \circ b = c$ where $\circ$ is a binary operation and "$a$", "$b$", "$\circ$", "$=$" and "$c$" are tokens. |
| Žunkovič & Ilievski (2022) | Perceptron Tensor Network | Rules-Based | 1D cellular automaton rule, 1D exponential and $D$-dimensional uniform ball. |
| Liu et al. (2022) | MLP Transformer | Algorithmic Image class. | Addition modulo $P$, regular addition and MNIST. |
| Nanda et al. (2023) | Transformer | Algorithmic | Addition modulo $P$. |
| Liu et al. (2023a) | MLP LSTM GCNN | Algorithmic Image class. Language Molecules | Regular addition, MNIST, IMDb dataset and QM9. |
| Merrill et al. (2023) | MLP | Algorithmic | Parity prediction and operations modulo prime. |
| Davies et al. (2023) | Transformer | Algorithmic | Addition modulo $P$. |
| Barak et al. (2022) | MLP Transformer PolyNet | Algorithmic | Parity prediction task. |
| Murty et al. (2023) | Transformer | Language | Question formation, tense-inflection and bracket nesting |
| Liu et al. (2023b) | MLP | Algorithmic | XOR, S4 group operation and bitwise XOR. |
| Varma et al. (2023) | Transformer | Algorithmic | Similar to Power et al. (2022) |

Table 1: Summary of datasets and architectures in which the grokking phenomenon has been studied. Note that *class.* is short for classification.

# B  Kolmogorov Complexity

The Kolmogorov complexity is one of several measures of complexity discussed in this paper. In some sense, it is the measure of complexity with the least prior knowledge used to generate its value.

**Definition B.1** (Kolmogorov). According to Yueksel et al. (2019), the Kolmogorov complexity $K_U$ of a string $x$ with respect to a universal computer $U$ is:

$$K_U(x) = \min_{p:U(p)=x} l(p) \tag{7}$$

where $p$ denotes a program and $l(p)$ is the length of the program in some standard language.

Essentially, the Kolmogorov complexity is the length of the minimal program required to produce a string representation of a model. To illustrate this point, consider the three strings presented in Figure. 7. Let us assume that each of the associated programs are minimal under some language understood by a universal computer $U^{10}$. In this case, the first string would have the least complexity as the program specifying it requires 52 characters, the second string would be more complex requiring 105 and the third would be most complex requiring 106. In the Kolmogorov formalism, if we have members $f, h \in \mathscr{F}$ and the minimal program to produce the string representation of $f$ is longer than $h$, it is more complex.

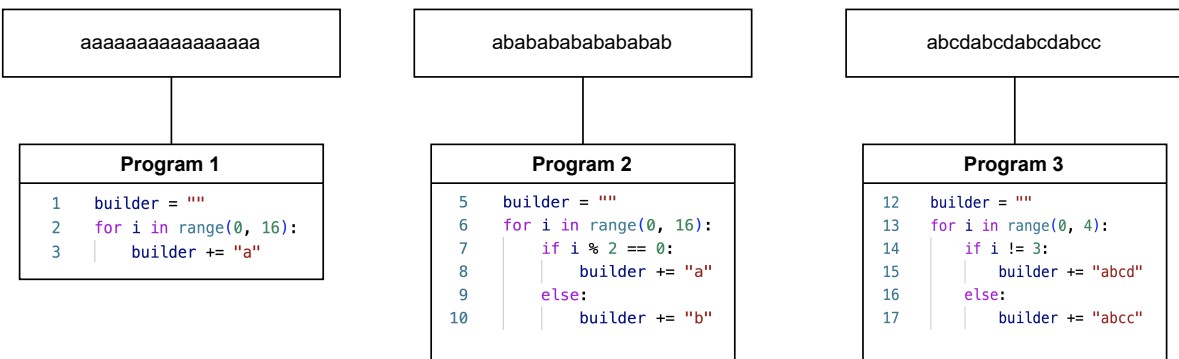

Figure 7: Illustration of Kolmogorov Complexity

With this Kolmogorov measure of complexity we can formalism the principle of parsimony under a computational picture. To do so, we use Solomonoff's theory of inductive inference. In Solomonoff induction, we assume that we have an observation about the world $o \in \mathscr{O}$ which can be encoded in a binary string. In addition, we have a set of hypothesis $\mathscr{H}$ about that observation. We assume that these hypothesises are computable in the sense that each can be run on a Turing machine producing $o$. Further, we make a metaphysical assumption that true hypothesises are generated randomly using an unbiased process whereby the binary sequence defining $h \in \mathscr{H}$ is generated by choosing between three options at every point in the sequence: 0, 1 or END. In this setting if a hypothesis $h$ generates an observation $o$ but has a smaller Kolmogorov complexity than other hypotheses generating $o$, it is more likely (Altair, 2012).

## B.1  Approximating Kolmogorov Complexity using Entropy

Unfortunately, the Kolmogorov complexity of a string (here a model) is non-computable (Vitányi, 2020). However, we can approximate it with a concept from information theory (Shannon, 1948). Namely, the entropy of a stochastic process which produces that string (or model).

**Theorem 1.** *Let a stochastic process $\{X_i\}$ be drawn i.i.d. according to the probability mass function $f(x)$, $x \in \mathcal{X}$, where $\mathcal{X}$ is a finite alphabet. According to Cover & Thomas (2006), we have the limit:*

$$\frac{1}{n}\mathbb{E}\left[K(X^n|n)\right] \to H(X). \tag{8}$$

---

[10]For example, we ignore the fact that we could simply print the strings using fewer character than these programs.

*I.e. the expected Kolmogorov complexity of a n-bit string approaches the entropy of the distribution from which characters in that string are drawn.*

Thus, if we treat each parameter of a model as a random variable, we may calculate the approximate Kolmogorov complexity of the model by considering the entropy of the distribution over the parameters which make up the model. For example, if we assume that each parameter $W$ in a particular model ($f$) is distributed normally, then the approximate[11] distribution (after quantisation) for $W$ is given in Equation 9 (Hinton & van Camp, 1993):

$$p(W) = \frac{t}{\sqrt{2\pi}} \exp\left[-\frac{W^2}{2}\right].$$ (9)

Now that we have a distribution over $W$, we can calculate $H(W)$:

$$
\begin{aligned}
H(W) &= \mathbb{E}_p\left[-\ln p(W)\right] \\
&= \mathbb{E}_p\left[-\ln\left(t\frac{1}{\sqrt{2\pi}}\exp\left[-\frac{W^2}{2}\right]\right)\right] \\
&= \frac{1}{2}\mathbb{E}_p[W^2] + \text{const.}
\end{aligned}
$$

Given this expression for $H(W)$, the approximate Kolmogorov complexity is then:

$$K(f) \approx \frac{N}{2}\mathbb{E}_W[W^2] + \text{const.}$$ (10)

Notably, $K(f)$ is proportional to the square of the weights of the network. Consequently, by minimising the approximate Kolmogorov complexity under the assumption of normality, we also minimise the $L_2$ norm of the weights. This is equivalent to the familiar weight decay regularisation strategy.

## B.2 Connection Between Kolmogorov Complexity and Bayesianism

We can also relate Kolmogorov complexity to Bayesian inference. Consider the case where we are finding parameters of a model using the *maximum a posteriori* estimator. In this paradigm, we wish to choose the model $f_\theta$ to maximise $P(f|D)$ where $D$ is the data. According to Vitányi & Li (2000), this is equivalent to finding $\theta^*$ such that:

$$
\begin{aligned}
f_\theta^* &= \min_\theta \left\{-\log P(D|f_\theta) - \log P(f_\theta) + \log P(D)\right\} \\
&= \min_\theta \left\{-\log P(D|f_\theta) - \log P(f_\theta)\right\}
\end{aligned}
$$ (11)

If we assume our hypothesis class to be finite and take the universal prior $m$, we can make the substitution $-\log P(f_\theta) = K(f_\theta)$ and $-\log P(D|f_\theta) = K(D|f_\theta)$ (Vitányi & Li, 2000). This gives:

$$f_\theta^* = \min_H \left\{K(f_\theta) + K(D|f_\theta)\right\}.$$ (12)

Hence, under the universal prior, the *maximum a posteriori* estimator is equivalent to the minimiser of the Kolmogorov complexity of the model and of the data given the model.

---

[11]Note that Equation 9 is in practice a very good approximation, since the quantisation $t$ is much smaller than the standard deviation of 1.

## C   Model Description Length

To understand model description length, we will consider two agents connected via a communication channel. One of these agents (Brian) is sending a model across the channel and the other (Oscar) is receiving the model. Both agree upon two items before communication. First, anything required to transmit the model with the parameters unspecified. For example, the software that implements the model, the training algorithm used to generate the model and the training examples (excluding the targets). Second, a prior distribution $P$ over parameters in the model . After initial agreement on these items, Brian learns a set of model parameters $\theta$ which are distributed according to $Q$. The complexity of the model found by Brian is then given by the cost of "describing" the model over the channel to Oscar. Via the "bits back" argument (Hinton & van Camp, 1993), this cost is:

$$\mathcal{L}(f_\theta) = D_{KL}(Q||P) \tag{13}$$

While the model description length is a relatively simple and quite general measure of model complexity, it relies upon a critical assumption. Namely, that the amount of information contained in the components agreed upon by Brian and Oscar should be small or shared among different models being compared. Fortunately, this is usually the case, especially in our experiments. When analysing the grokking phenomenon, we generally agree upon priors and look at the complexity of a model across epochs in optimisation. Thus, there is complete shared information cost outside of changes which occur during optimisation.

The model description length is often combined with a data fit term under the minimal description length principle or MDL (MacKay, 2003). One can understand this principle by considering the following scenario. Oscar does not know the training targets $Y$ but would like to infer them from $X$ to which he has access. To help Oscar, Brian transmits parameters $\theta$ which Oscar then uses to run $f_\theta$ on $X$. However, the model is not perfect; so Brian must additionally send corrections to the model outputs. The combined message of the corrections and model, denoted $(D, f)$, has a total description length of:

$$\mathcal{L}(D, f_\theta) = \mathcal{L}(f_\theta) + \mathcal{L}(D|f_\theta). \tag{14}$$

In reference to Equation 1, the model description length is taking the role of the complexity term and the residuals are taking on the role of the data fit. The model which minimises $\mathcal{L}(D, f_\theta)$ is deemed optimal under the MDL principle (Hinton & van Camp, 1993) and also under our generalised objective function in Equation 1.

Notably, the MDL principle is equivalent to two widely used paradigms for model selection. The first is MAP estimation from Bayesian inference where $\mathcal{L}(f_\theta)$ comes to represent a prior over the parameters which specify $f_\theta$ (MacKay, 2003). Alternatively, if one calculates $L(D, f_\theta)$ under a Gaussian prior and posterior where the standard deviation of these distributions are fixed in advance, the model complexity term reduces to a squared weight penalty and the data fit is proportional to the mean squared error (Hinton & van Camp, 1993)[12].

---

[12]See Equation 4 of Hinton & van Camp (1993)

## D  A note on GP Complexity

While the model description length is equivalent to many complexity measures used in model selection, it is not equivalent to all. For example, we complete some experimentation with GP regression where we observe grokking across optimisation of the kernel hyperparameters. The posterior distribution of these kernel parameters is given by:

$$p(\theta|X, y) = \frac{p(y|X, \theta)p(\theta)}{p(y|X)} = \frac{p(\theta) \int p(y|f, X)p(f|\theta)df}{p(y|X)} \tag{15}$$

From here we could use MAP-estimation to find $\theta$ which we know is equivalent to the MDL. However, it is standard practice in GP optimisation to find $\theta$ which maximises $p(y|X, \theta)$ (Rasmussen & Williams, 2006). It turns out $p(y|X, \theta)$ itself contains a regularising term which is often labelled the complexity[13] (Rasmussen & Williams, 2006):

$$\log p(y|X, \theta) = -\underbrace{\frac{1}{2}y^T K_\theta^{-1} y}_{\text{data fit}} - \underbrace{\frac{1}{2}\log |K_\theta|}_{\text{complexity}} - \underbrace{\frac{n}{2}\log 2\pi}_{\text{normalisation}} \quad . \tag{16}$$

In Equation 16, this so-called complexity penalty characterises "the volume of possible datasets that are compatible with the data fit term" (Bauer et al., 2016). This is clearly distinct from the model description length. However, Equation 16 is still congruent with Equation 1 and thus amenable to analysis in this paper. As we will see, even though this definition of complexity is not equivalent to the description length, it seems to serve the same function empirically and is treated in the same way by the mechanism we propose for grokking.

---

[13]Note that in the equation, $K_\theta = K_f + \sigma_n^2 I$, which is the covariance function for targets with Gaussian noise of variance $\sigma_n^2$

# E    Datasets used in Experimentation

Many datasets were used for the experimentation completed in this paper. They are were either found in Merrill et al. (2023), Power et al. (2022) or were developed independently. Although grokking has been seen on non-algorithmic datasets (Liu et al., 2023a), we restrict ourselves to these and a basic regression task since our focus is a theoretical exploration of the phenomenon via the modification of other inducing variables. Work on larger datasets may have hindered this exploration.

**Dataset 1** (Parity Prediction Task)**.** *In the parity prediction task, the model is provided with a binary sequence $x$ of length $k$. The target $y$ is the parity of the sequence i.e. the product of the sequence if $0$ is $-1$ and $1$ is $1$.*

**Dataset 2** (Prime Modulo Addition Task)**.** *In the prime modulo addition task, a prime $p$ and two numbers in the range $[0, p)$ are chosen. These numbers are represented by a one hot encoding and the model must predict their addition modulo $p$.*

**Dataset 3** (Prime Modulo Subtraction Task)**.** *The prime modulo subtraction task has the same setup as Dataset 2, but is subtraction.*

**Dataset 4** (Prime Modulo Division Task)**.** *The prime modulo division task has the same setup as Dataset 2, but is division.*

**Dataset 5** (Prime Modulo Polynomial Task)**.** *The prime polynomial division task has the same setup as Dataset 2, but the model tries to predict the result of the equation:*

$$x \circ y = x^2 + xy + y^2 \mod p \tag{17}$$

**Dataset 6** (Extended Prime Modulo Polynomial Task)**.** *The extended prime polynomial division task has the same setup as Dataset 2, but the model tries to predict the result of the equation:*

$$x \circ y = x^2 + xy + y^2 + x \mod p \tag{18}$$

**Dataset 7** (Extended Prime Modulo Multiplication Task)**.** *The extended prime polynomial division task has the same setup as Dataset 2, but the model tries to predict the result of the equation:*

$$x \circ y = x \cdot y \cdot x \mod p \tag{19}$$

**Dataset 8** (1-0 Classification)**.** *In this classification task, a model is attempting to distinguish whether a point will take a value of $0$ or $1$. These points are normally distributed (with $\sigma = 1$) around $0$ and have label $y = 0$ if they are below $x = 0$ and $y = 1$ if they are above.*

**Dataset 9** (1-0 Classification on a Slope)**.** *In this classification task, a model is attempting to distinguish whether a point will be above $0$ or not. The $x$ values of these points are normally distributed (with $\sigma = 1$) around $0$ and the $y$ values are given by the linear equation:*

$$y = 0.3x \tag{20}$$

**Dataset 10** (Regression on a Sine Wave)**.** *In this regression task, a model is attempting to predict the value of the equation:*

$$y = A\sin(2\pi f x + \phi) + B\epsilon \tag{21}$$

*where by default $A = 1$, $f = \frac{1}{\pi}$, $\phi = 0$, $B = 0.1$ and $\epsilon$ is noise distributed according to $\mathcal{N}(0, 1)$. Further $x$ values in the training set are modified so that the function is not necessarily on a uniform support by adding Gaussian noise of the same form as $\epsilon$.*

## F   Statistical Analysis of the Concealment Strategy

### F.1   Regression Method

Given a matrix $X$ of pairs $(l, \delta)$ where $l$ is the additional length and $\delta$ is the recorded grokking gap, we first transform the $y$-values into log space. That is, the dataset of pairs is given by:

$$X' = \begin{bmatrix} X_0 & \ln X_1 \end{bmatrix} \tag{22}$$

Then we find the optimal coefficients $a$ and $b$ such that the model:

$$\hat{y} = aX_0' + b \tag{23}$$

has minimal squared error with respect to the labels $X_1'$. Returning from log to regular space, the function:

$$\delta = \exp\left(al + b\right) \tag{24}$$

is then our proposed relationship between additional dimensionality and grokking gap.

### F.2   Correlation and $p$-Value Results

| Dataset | $r$ | $p$ |
|---------|-----|-----|
| Combined | 0.92 | $1.46 \cdot 10^{-37}$ |
| Addition | 0.933 | $3.91 \cdot 10^{-7}$ |
| Subtraction | 0.914 | $1.84 \cdot 10^{-6}$ |
| Division | 0.967 | $4.659 \cdot 10^{-9}$ |
| Polynomial | 0.959 | $1.860 \cdot 10^{-8}$ |
| Extended Polynomial | 0.906 | $2.981 \cdot 10^{-6}$ |
| Extended Multiplication | 0.942 | $1.484 \cdot 10^{-7}$ |

Table 2: Table summarising the Person correlation $r$ and null hypothesis likelihood $p$ for concealment data in log space. Note that all datasets are under a modulo argument.

# G   Connection Between the Complexity Theory of Grokking and Previous Theories

The complexity theory of grokking unifies loss and representation based theories of grokking under a single framework. In the text below, we show how each may be seen as an example of the behaviour described in Mechanism 1.

The loss-based explanation of Liu et al. (2023a) asserts that in the weight space of a model there exists a Goldilocks zone of high generalisation. In cases of grokking, models will quickly find over-fitting solutions before being guided towards this Goldilocks zone via weight decay. As previously mentioned, weight norm can be seen as a measure of a model's description length. Thus the guidance of weight norm is towards regions of lower complexity. Additionally, via Assumption 1, we can say that the Goldilocks zone must be a region of LELC since good generalisation comes from lower complexity. Consequently, this theory can be viewed as an instance of our broader framework with over-fitting solutions constituting LEHC and the Goldilocks zone constituting LELC.

The use of the LMN by Liu et al. (2023b) is also congruent with our theory of grokking. In the paper, the authors introduce LMN as a complexity measure and show that a decrease in LMN after a period of high training performance leads to a grokking solution. Under Mechanism 1, LMN is one instance of a complexity measure which can result in grokking.

Theories which use representation dynamics as a means of explaining grokking can also be placed in our framework. Via Assumption 1, more general representations should have a relatively low complexity. Thus, representation descriptions which talk of an emergence of general structures after the initial creation of memorising structures are talking of a transition from LEHC to LELC. Indeed, in neural networks it may be the case that LEHC solutions can often be characterised as "memorising" and LELC as "general circuits." However, in other models we require the more abstract language of Mechanism 1.

## H   Additional Model Experiments

### H.1   Linear Regression with a Specific Seed

In the following experiment, we take the same learning setting as in Section 2.1.1 but restrict ourselves to one seed which shows a clear case of grokking. This case is presented in Figure 8

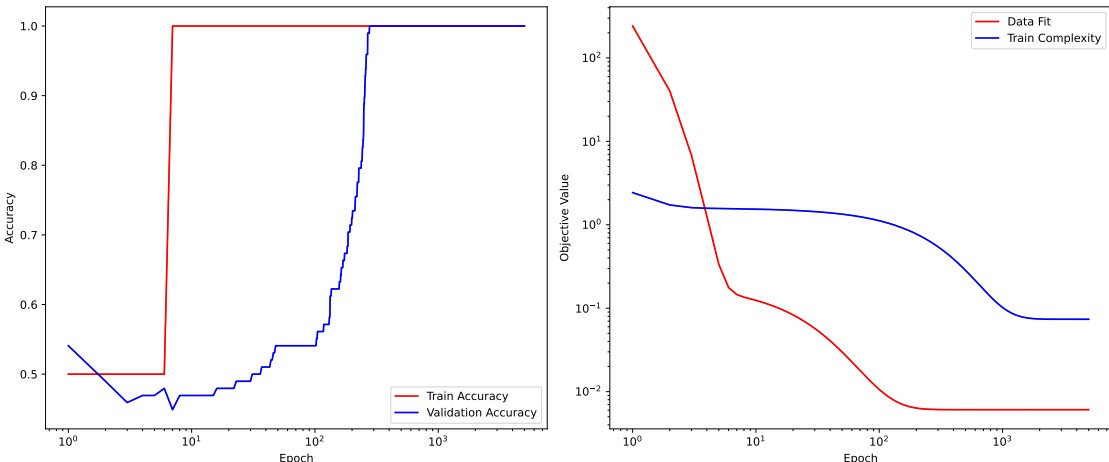

Figure 8: Accuracy, complexity and data fit on zero-one slope classification task with a linear model and one seed.

### H.2   GP Classification Complexity for 0-1 Classification

We take the same learning setting as in Section 2.1.2, showing the complexity and error curves in Figure 9. In this figure, the data fit term is the negation of the error in Equation 6. Alternatively, the complexity is measured as the KL divergence in Equation 6. That is, the data fit and complexity are:

$$\text{data fit} = -\sum_{i=1}^{N} \mathbb{E}_{q_\phi(f_i)}[\log p(y_i|f_i)], \quad \text{complexity} = \text{KL}[q_\phi(f)||p_\theta(f)] \tag{25}$$

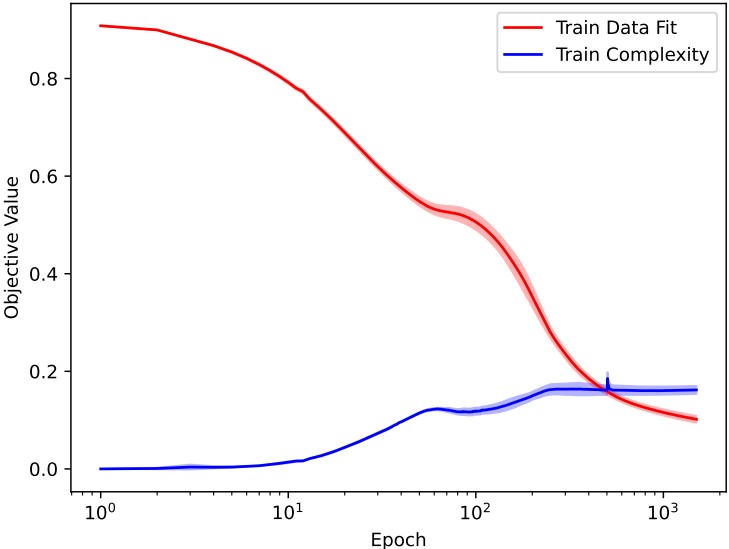

Figure 9: Complexity and data fit for GP classification on the 0-1 task.

## H.3 GP Classification Complexity for Algorithmic Case

We take the same learning setting as in Section 2.1.2, showing the complexity and error curves in Figure 9 as defined in Equation 25.

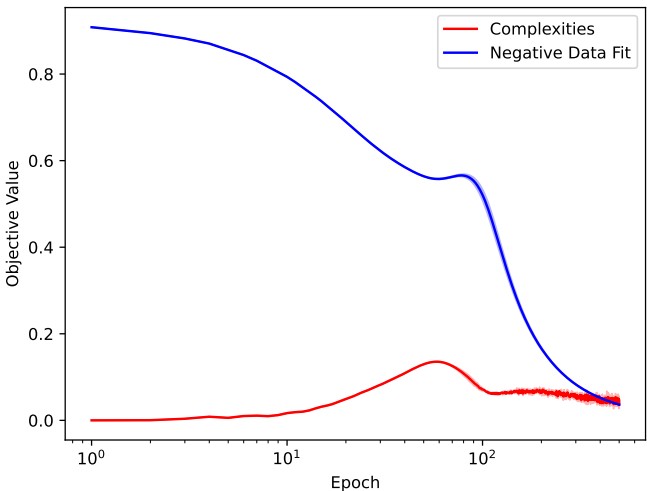

Figure 10: Complexity and data fit for GP classification on the concealed parity prediction task.

# I Grokking via Concealment

## I.1 Algorithm

---

**Algorithm 1** Algorithm for relationship of $\Delta_k$ with dimensionality

---

**Require:** $|L| > 0$                                               ▷ Number of additional lengths
**Require:** $|D| > 0$                                                     ▷ Number of datasets
**Require:** $|R| > 0$                                               ▷ Number of random seeds
 1: $\gamma \leftarrow 0.95$                                                  ▷ Threshold for high accuracy
 2: $N \leftarrow 1500$                                                  ▷ Number of epochs
 3: `array_of_avg, array_of_std` $\leftarrow [], []$
 4: **for** each $l$ in $L$ **do**
 5:     `length_scale_avgs, length_scale_stds` $\leftarrow [], []$
 6:     **for** each $d$ in $D$ **do**
 7:         `dataset_gap` $\leftarrow []$
 8:         **for** each $r$ in $R$ **do**
 9:             `set_random_seed`$(r)$
10:             `model` $\leftarrow$ `initialise_model`$()$
11:             `training_accuracy, validation_accuracy` $\leftarrow$ `train_model`$(\text{model}, d)$
12:             `training_index` $\leftarrow$ `first_index_above`$(\text{training\_accuracy}, \gamma)$
13:             `validation_index` $\leftarrow$ `first_index_above`$(\text{validation\_accuracy}, \gamma)$
14:             $\Delta_k \leftarrow$ `validation_index` $-$ `training_index`
15:             `dataset_gap`.append$(\Delta_k)$
16:         **end for**
17:         `avg_gap, std_gap` $\leftarrow$ avg(`dataset_gap`), std(`dataset_gap`)
18:         `length_scale_avgs`.append(`avg_gap`)
19:         `length_scale_stds`.append(`std_gap`)
20:     **end for**
21:     `array_of_avg`.append(`length_scale_avgs`)
22:     `array_of_std`.append(`length_scale_stds`)
23: **end for**

---

## I.2 Original Regression Plot

Below we present a version of Figure 4 where we have reversed alterations to the error bars and included data from the 0 additional length category.

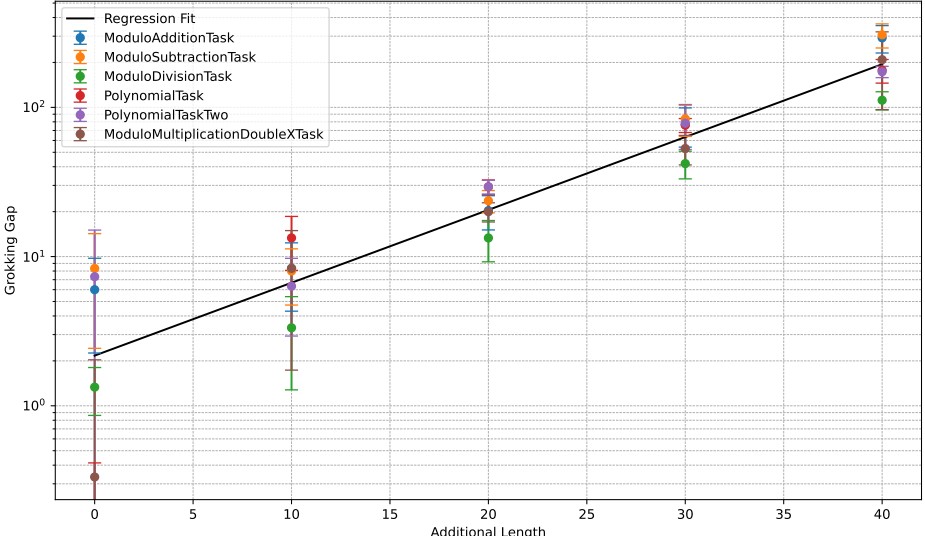

Figure 11: Relationship between grokking gap and number of additional dimensions using the grokking via concealment strategy.

## J Illustrative Figures

### J.1 Lengthscale Plot of GP with Parity Prediction

In Section 2.1.2, a GP is used for the hidden parity prediction task. With this dataset, models should learn to disregard dimensions with spurious data. In this case, the spurious dimensions are those above three. Presented in Figure 12 is a plot of the length scale parameter over the course of hyperparameters. As can be seen in that figure, length scale increases for for input dimensions above three but decrease for input dimensions three and below. This is expected – a larger length scale cannot discern small changes in the input dimension and thus renders them uninformative.

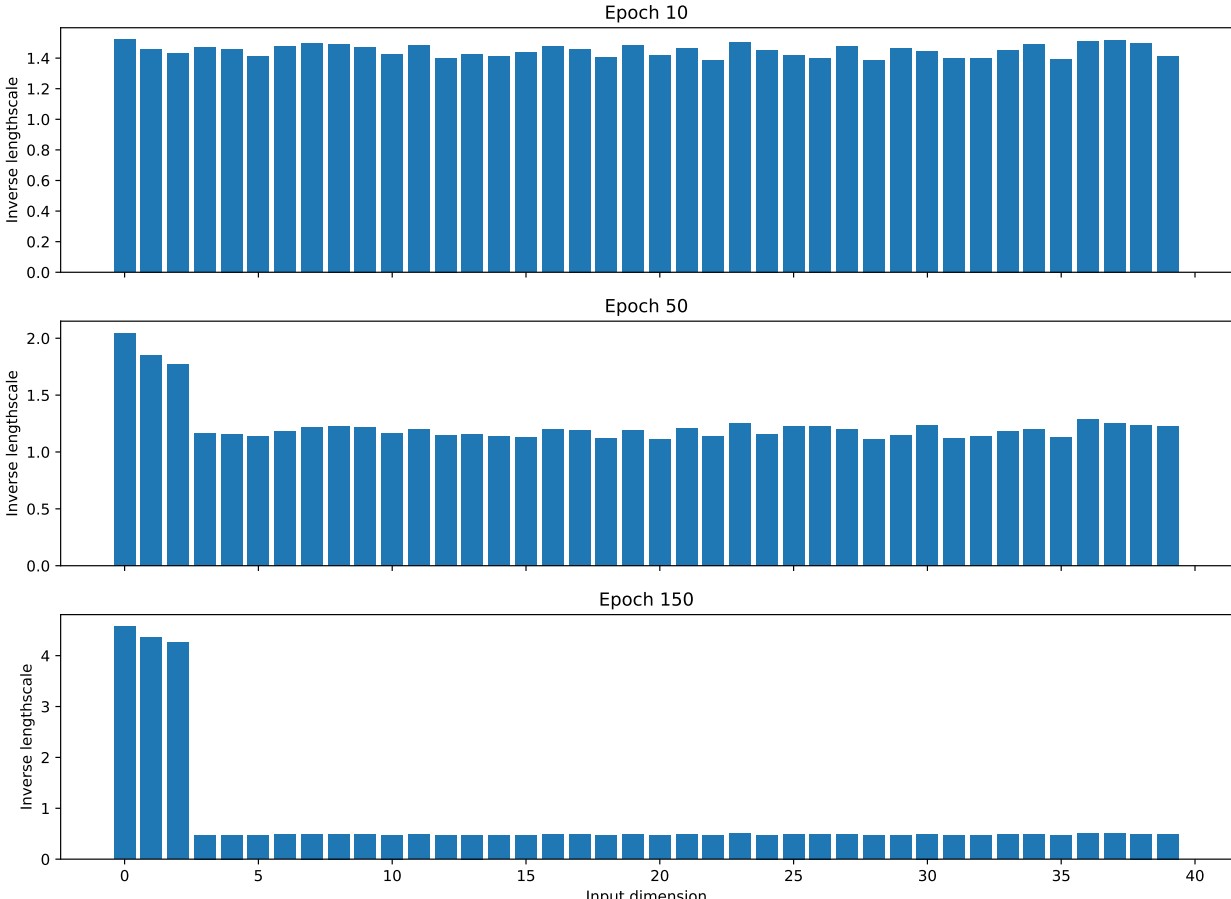

Figure 12: Example of GP kernel length scale for hidden parity prediction task. Note that inverse length scale refers to $1/l_i$ where $l_i$ is the length scale associated with dimension $i$.

## J.2 Illustrative Example of Grokking

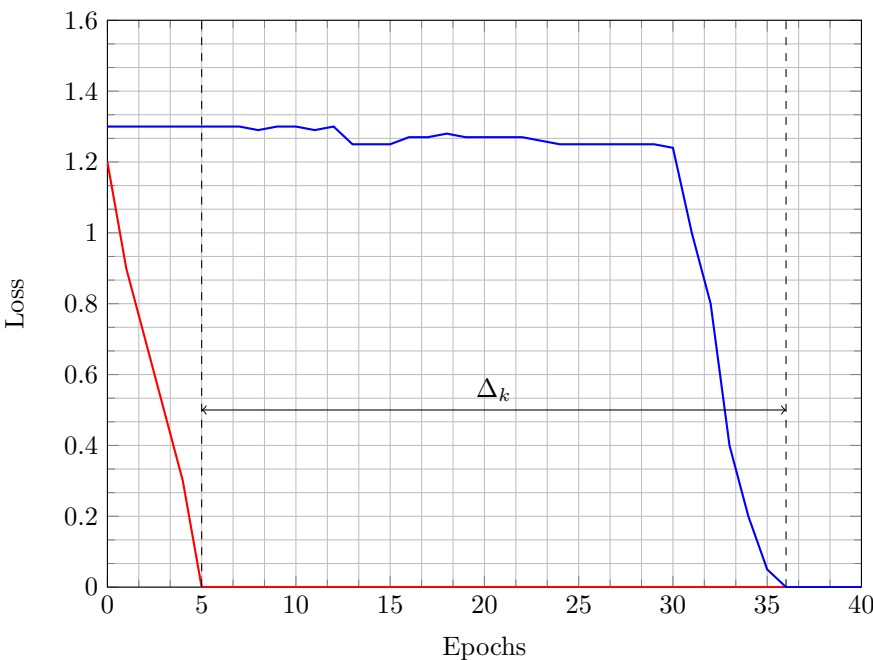

Figure 13: Illustrative example of the grokking phenomenon. The red line is the training loss and the blue line is the validation loss.

# K    Further Analysis of Linear Classification

In this appendix we provide a further analysis of the linear classification grokking case covered in Section 2.1.1. Specifically, we examine the congruency of these results with our grokking hypothesis (Section 3). To do so, we conducted three further experiments. In the first we alter the way we initialise the model and consider the trend between the grokking gap, complexity gap and data fit gap (defined in the relevant subsection). In the second experiment, we examine the evolution of weights in the linear model over the course of optimisation. Finally, in the third experiment we remove the weight penalty altogether and find that this removes grokking altogether.

## K.1    Altering the Initialisation of the Linear Model

In this experiment we considered the trend between three variables whilst altering the model's initialisation[14]. Specifically we changed the value of $\alpha$ in the following equation:

$$w_0 = \begin{bmatrix} \alpha & 1 - \alpha & 1 - \alpha & 1 - \alpha \end{bmatrix} \tag{26}$$

The three variables of interest are:

- The grokking gap (defined in the introduction)

- The complexity gap. I.e. the difference in complexity between the point that the model does well on the training set when compared to the validation set.

- The data fit gap. I.e. the difference in the data fit term between the point that the model does well on the training set when compared to the validation set.

Under the grokking mechanism we propose, we would expect these variables to be related in the following ways. Firstly, the grokking gap should be a decreasing function with the complexity gap since the required time to go from a region of HELC to LELC determines the gap width. Secondly, the data fit gap should be increasing with initialisation weight since in these cases we see a smaller grokking gap with learning driven primarily by the data fit rather than complexity (we are in a region of HELC). In Figure 14, one can see that these predicted trends are supported when one alters $\alpha$ in Equation 26.

## K.2    Examining the Evolution of Weights in the Linear Model

We also considered the evolution of weights in the linear model. Specifically, we experimented with different values of $\alpha$ in Equation 26, examining the 4 weights of the linear regression model in each case (all values were averaged across five random seeds). The results of this experimentation are in Figure 15. We found that a similar solution was reached no matter the initialisation employed. In addition, the three "distracting" values of the input are slowly "ignored" over the course of training. This corresponds to a lower complexity solution under the $L_2$ norm employed for the regression.

---

[14]Employing five random seeds which are averaged to produce Figure. 14.

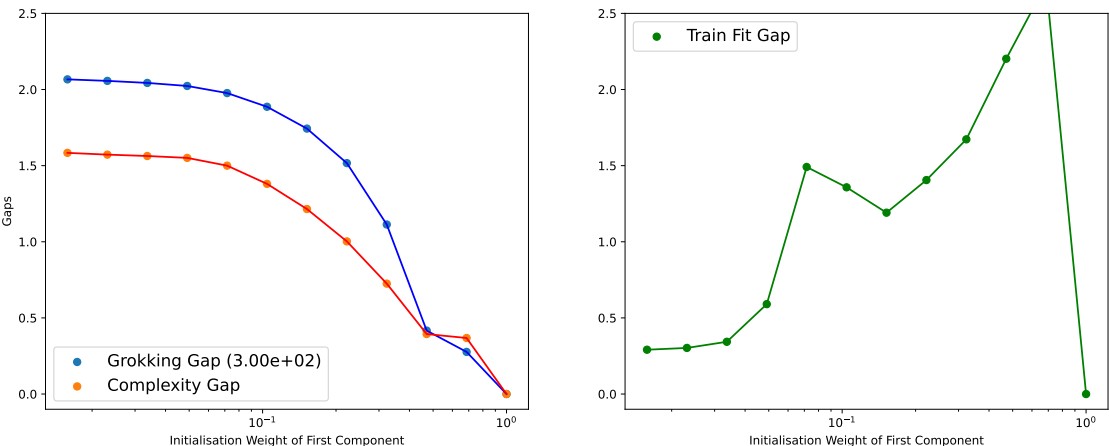

Figure 14: Further initialisation analysis of the model presented in Section 2.1.1. Gaps refers to the particular gap (grokking, complexity or data fit) which is presented in the graph. The $x$-axis is the initialisation weight of the first component i.e. $\alpha$ in Equation 26. Note that all gaps are approximately 0 when $\alpha$ is 1 since the solution to the problem given by the first component of the input. The number in parentheses after "Grokking Gap" refers to the value points on the graph are divided by.

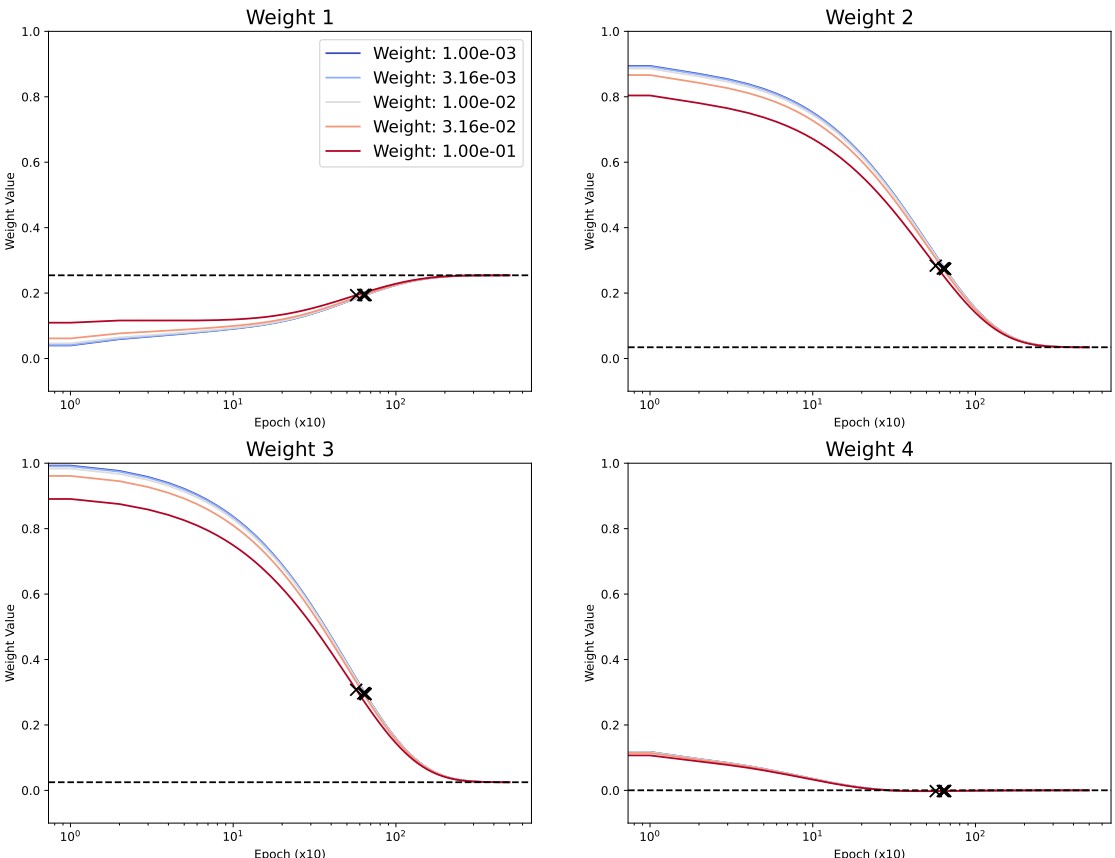

Figure 15: Evolution of weights in the linear model. Black crosses correspond to the values of the weights at which grokking occurred.

### K.3 Removing Weight Decay

If we remove the weight penalty term used for optimisation of the linear model, we do not see grokking even after 1,000,000 epochs of optimisation. The lack of grokking can be seen in Figure 16. Note that only every 100th point is plotted.

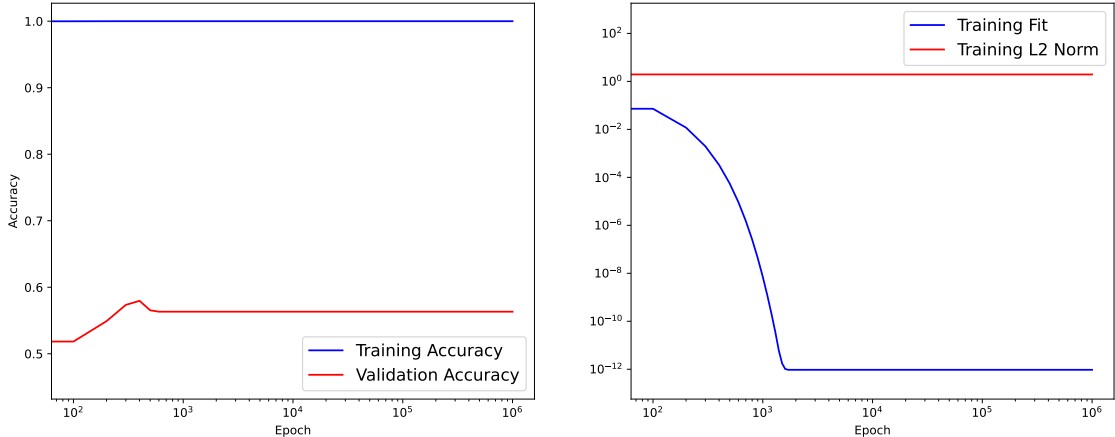

Figure 16: Accuracy, data fit and complexity on zero-one slope classification task with a linear model and no weight penalty. Note that the shaded region corresponds to the standard error of five training runs.

## L   Further Analysis of GP Classification

To determine the necessity of a complexity regularisation term for grokking in GP classification, we removed it from the training of models under learning scenarios 2 and 3 (Section 2.1.2). As can be seen in Figures 17 and 18. Note that for both figures, only every 5th point is plotted.

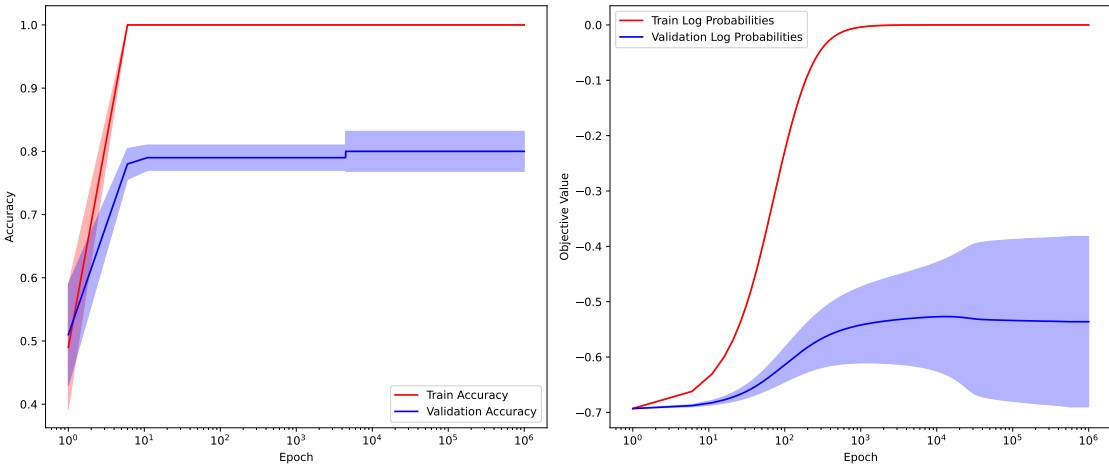

Figure 17: Accuracy and log likelihoods of RBF GP on the 0-1 prediction task without added complexity term. Note that the shaded region corresponds to the standard error of five training runs. *Acc.* is *Accuracy* and *Val.* is *Validation.*

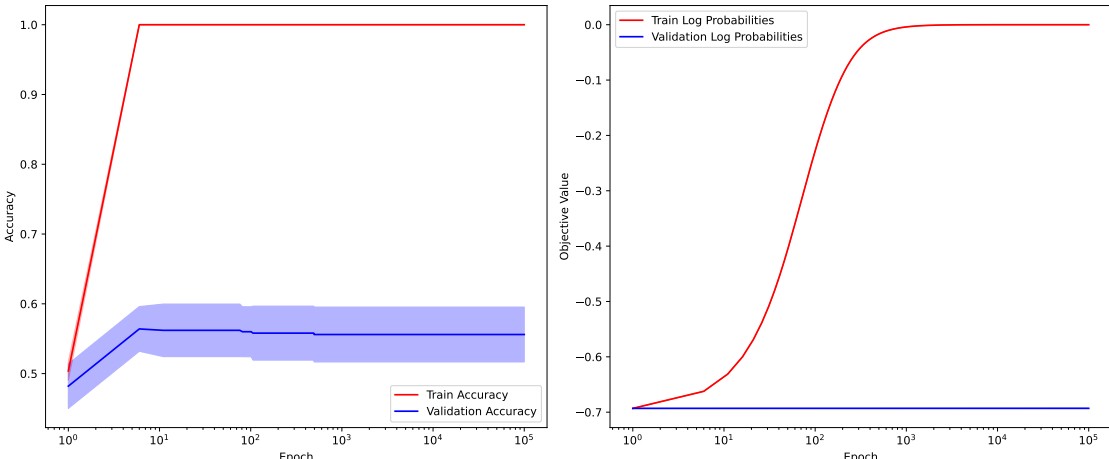

Figure 18: Accuracy and log likelihoods of RBF GP on the hidden parity prediction task without added complexity term. Note that the shaded region corresponds to the standard error of five training runs. *Acc.* is *Accuracy* and *Val.* is *Validation.*

## M Further Analysis of Concealment Data Augmentation

We also considered how the magnitude of initial weights influenced the concealment data augmentation strategy. Under our theory, smaller initial weight magnitudes should decrease the propensity for grokking since models start in a region of lower complexity. We would predict this trend should, however, be counterbalanced by introducing additional spurious dimensions in the input space. Thus, all models should exhibit an increase in grokking with additional spurious dimensions but those models with smaller magnitudes should exhibit a smaller increase. Indeed, this is what we see in Figure 19, where we plot the *relative grokking gap* since models with smaller initial weights took longer to fit the training data. Note that the definition of relative grokking gap $\tilde{\Delta}_k$ is:

$$\tilde{\Delta}_k = \frac{\Delta_k}{E_1} \tag{27}$$

where $E_1$ is the epoch at which the model reached high training performance.

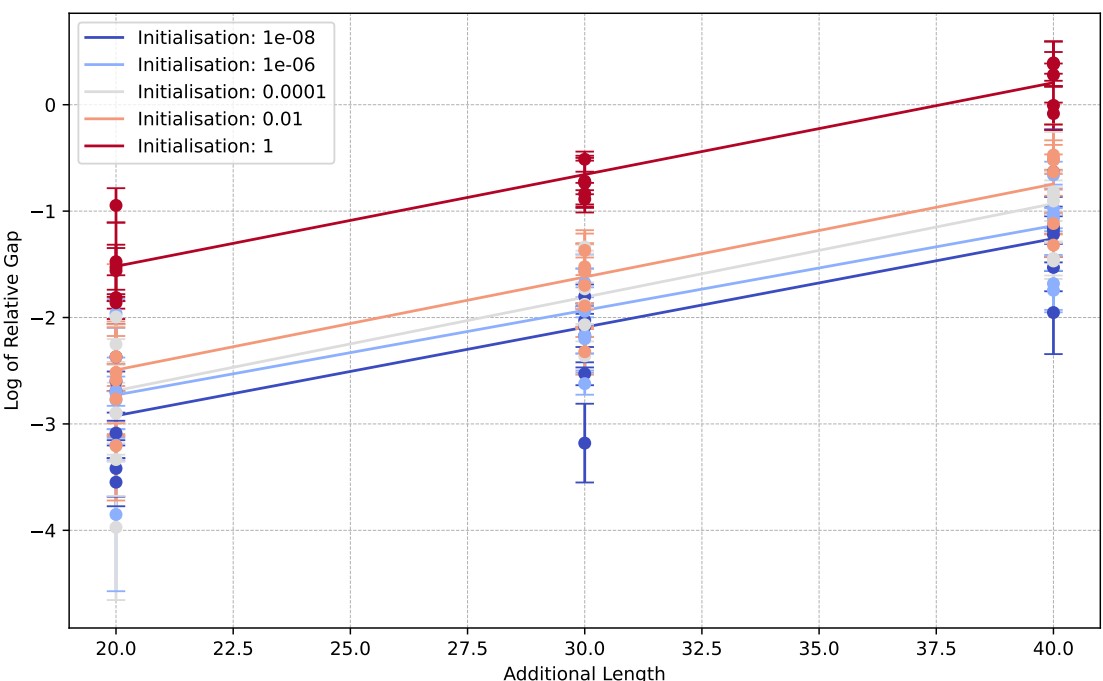

Figure 19: Relationship between relative grokking gap and number of additional dimensions using the grokking via concealment strategy. Datasets used were the same as in Figure 4.

## N   Loss plots from grokking with concealment

In Figure 20 we show a collection of loss plots from our experiments inducing grokking via concealment. Each of the datasets can again be found in Appendix E. The number of spurious dimensions used for these particular plots is $k = 30$. The model is trained on each task for 500 epochs.

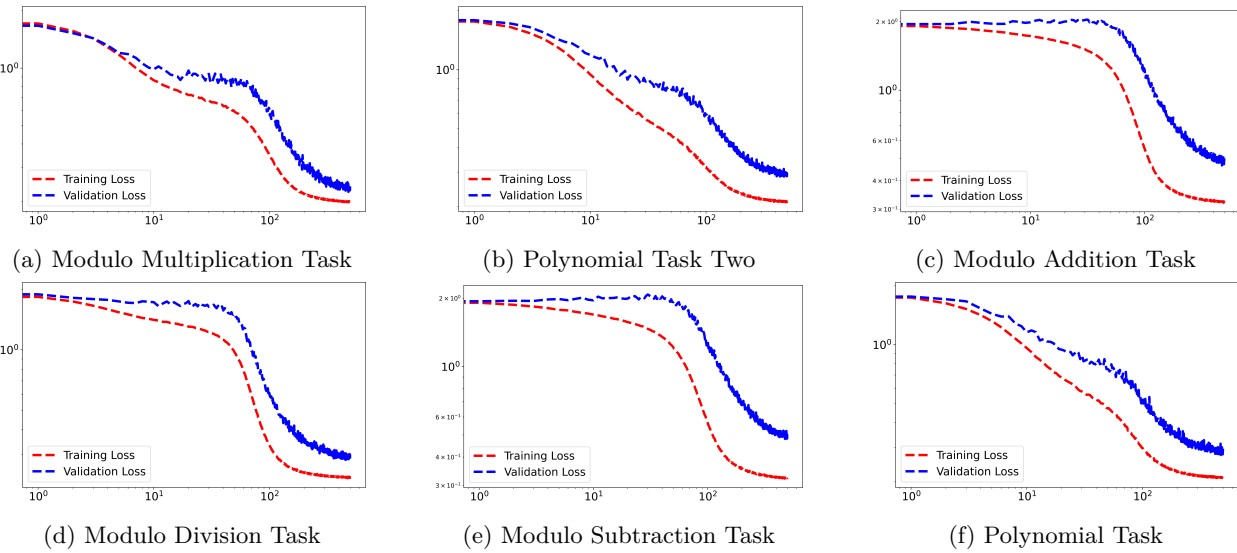

(a) Modulo Multiplication Task        (b) Polynomial Task Two        (c) Modulo Addition Task

(d) Modulo Division Task        (e) Modulo Subtraction Task        (f) Polynomial Task

Figure 20: Loss plots from all experiments when inducing grokking via concealment with 30 spurious dimensions.

## O   On the complexity term in approximate inference for GP models

As shown in the main text, the exact log marginal likelihood (LML) for GP regression has interpretable terms as follows:

$$\mathcal{L}(\theta) = \log p(y|X, \theta) = -\underbrace{\frac{1}{2} y^T K_\theta^{-1} y}_{\text{data fit}} - \underbrace{\frac{1}{2} \log |K_\theta|}_{\text{complexity}} - \underbrace{\frac{n}{2} \log 2\pi}_{\text{normalisation}} \quad . \tag{28}$$

where $K_\theta = K_f + \sigma_n^2 I$. Unfortunately, the exact LML is analytically intractable for many other GP models of interest such as GP classification. In the following, we will write down the approximate LML provided by variational inference or the Laplace's approximation. We will then look at their special case - GP regression, to identify which terms correspond to the data fit term and the complexity term in the GP regression's LML.

### O.1   Laplace approximation

Laplace's method approximates the posterior by a Gaussian density where its mean is the mode of the posterior and its covariance is the inverse of the negative Hessian evaluated at the mode. The corresponding approximate LML is

$$\mathcal{F}_{\text{Laplace}}(\theta) = -\underbrace{\frac{1}{2} \hat{\mathbf{f}}^T K_\theta^{-1} \hat{\mathbf{f}} + \log p(y|\hat{\mathbf{f}})}_{\text{data fit}} - \underbrace{\frac{1}{2} \log |B|}_{\text{complexity}}, \tag{29}$$

where $\hat{\mathbf{f}}$ is the posterior mode, $B = I_n + W^{\frac{1}{2}} K W^{\frac{1}{2}}$, and $W = -\nabla\nabla \log p(y|\mathbf{f})$. For GP classification with a logistic likelihood function, the second derivative of the likelihood function wrt the function value does not depend on the target. That is, $B$ does not depend on y. For this reason, we can view $\frac{1}{2} \log |B|$ as a model complexity measure. To double check, when the likelihood is Gaussian as in GP regression, $B = \frac{1}{\sigma_n^2}(K_{\mathbf{f}} + \sigma_n^2 I_n)$ and hence $\frac{1}{2} \log |B| = \frac{1}{2} \log |K_{\mathbf{f}} + \sigma_n^2 I_n| - \frac{N}{2} \log \sigma_n^2$, which is identical to the GPR's complexity term up to a constant when the noise is fixed.

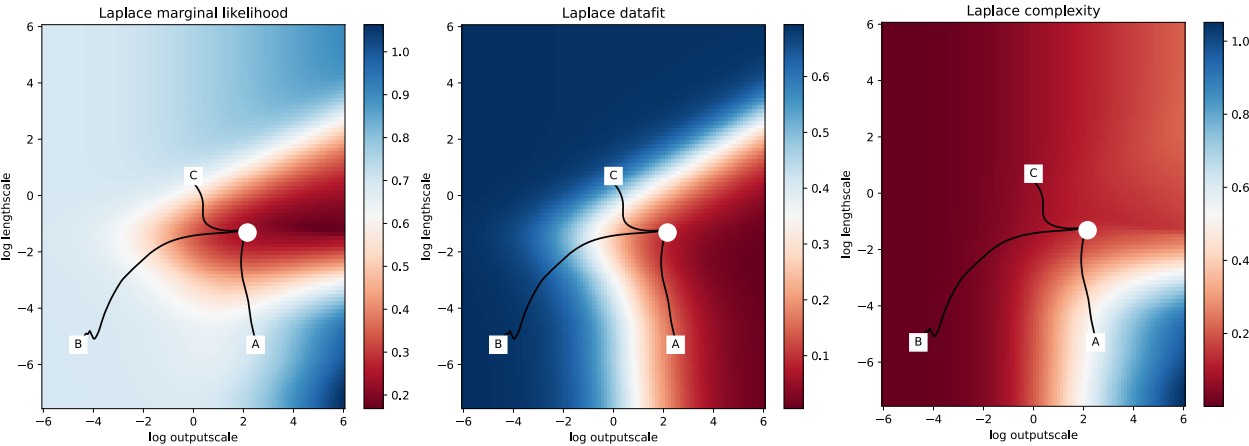

Figure 21: The approximate log marginal likelihood, data-fit and complexity provided by the Laplace's approximation. The black curves are the trajectories of the hyperparameters with different initialisations when using variational inference.

### O.2   Variational inference and the lower bound on the LML

Gaussian variational inference is widely used to perform approximate inference in GP models with non-Gaussian likelihoods such as GP classification. In this setting, the variational lower bound on the LML can be optimised to select the variational approximation and a point estimate for the model hyperparameters.

Assuming $p(\mathbf{f}) = \mathcal{N}(\mathbf{f}; 0, K_\mathbf{f})$ and $q(\mathbf{f}) = \mathcal{N}(\mathbf{f}; \mu, \Sigma)$, the variational bound can be written as follows,

$$\mathcal{F}_{\text{VI}}(q(\mathbf{f}), \theta) = \underbrace{-\text{KL}(q(\mathbf{f})||p(\mathbf{f}))}_{\text{KL term}} + \underbrace{\int_\mathbf{f} q(\mathbf{f}) \log p(y|\mathbf{f})}_{\text{expected log likelihood}} \tag{30}$$

$$= -\frac{1}{2}\mu^\intercal K_\mathbf{f}^{-1}\mu - \frac{1}{2}\text{trace}(K_\mathbf{f}^{-1}\Sigma) - \underbrace{\frac{1}{2}\log|K_\mathbf{f}|}_{\text{prior entropy+const}} + \frac{1}{2}\log|\Sigma| - \frac{N}{2} + \int_\mathbf{f} q(\mathbf{f}) \log p(y|\mathbf{f}) \tag{31}$$

Using the KL term as a complexity measure in this case has two connected issues:

- Unlike exact GP regression or Bayesian neural networks with a fixed prior, both the variational approximation and the hyperparameters are being optimised in variational inference. It is thus challenging to state which terms in the KL correspond to the complexity of the current fit, and which corresponds to the complexity governed by the prior model. In fact, for the GP regression case and $q(\mathbf{f})$ is set to the exact posterior, the KL term has a data-fit component and thus does not naturally fall back to the complexity term in the GPR's exact LML.

- One could pick the variational posterior to be the prior (which is of course a poor fit), leading to a zero KL term. However, this does not mean the complexity is zero!

Due to these potential issues, we did not see a clear relationship between the KL term and grokking in GP classification, as shown in the main text and Sections H.2 and H.3. We can work around these issues by looking at just the entropy of the prior (which resembles the complexity term in the GPR LML) or using the complexity term provided by Laplace's method using the current hyperparameter estimates. Figure 22 shows the objective function, the learning curves and predictions made during training, for various hyperparameter initialisations. The initialisations and trajectories are shown together with the Laplace approximate LML in in Figure 21.

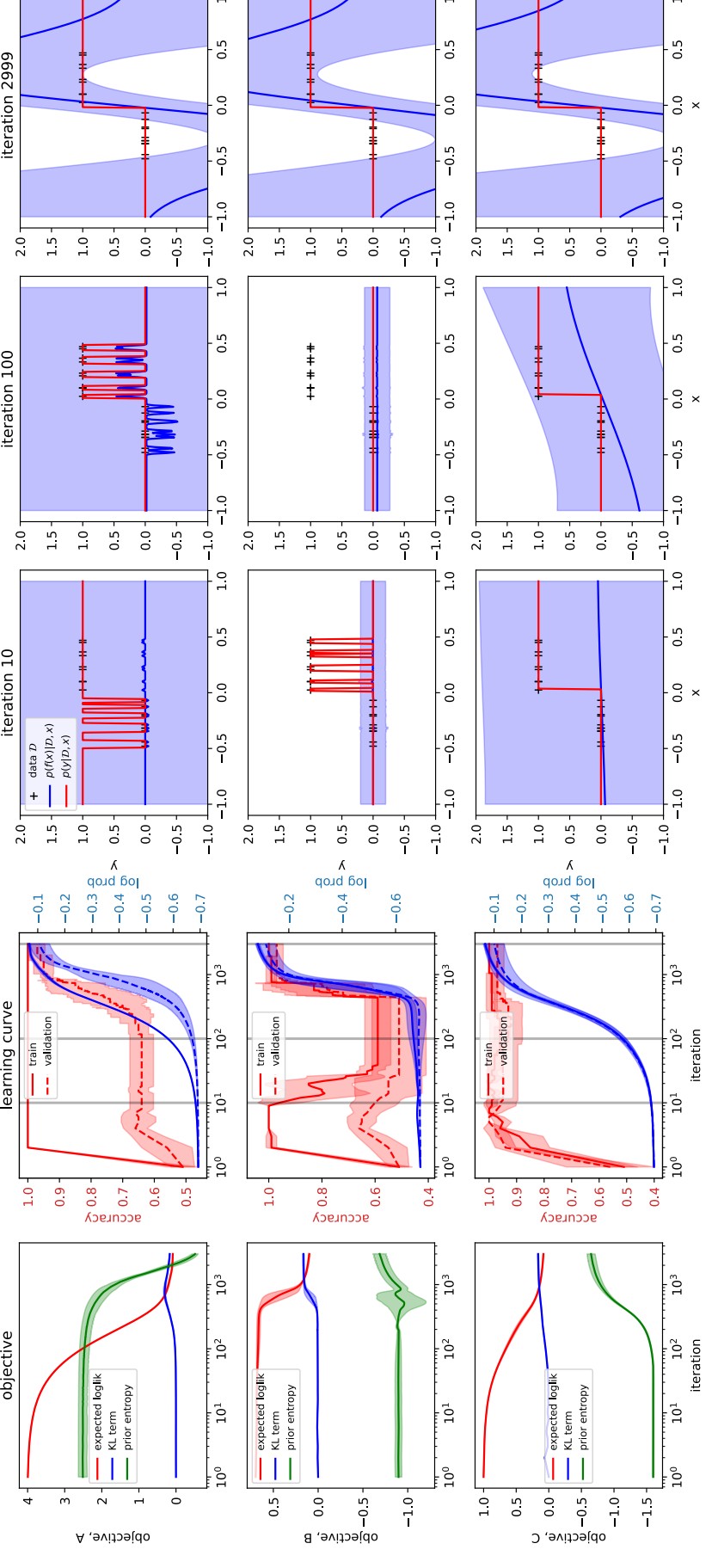

Figure 22: The objective function [first column], the learning curves [second column] and predictions made during training [final three columns], for three hyperparameter initialisations [each row corresponds to an initialisation]. A: short lengthscale and high kernel variance. B: short lengthscale and small kernel variance. C: intermediate lengthscale and kernel variance.

