# OpenReview forum: "Grokking Beyond Neural Networks: An Empirical Exploration with Model Complexity"
_TMLR — Accepted by TMLR_

### Review · Reviewer_eZfL · 2023-11-22

**Summary Of Contributions:**

This paper studies a phenomenon known as *grokking*, where they achieve perfect or near-perfect accuracy on the validation set long after the same performance has been achieved on the training set.

**Audience:**

Yes

**Claims And Evidence:**

Yes

**Requested Changes:**

1. Could the authors explain the high-level idea behind why Grokking occurs in both neural networks and Gaussian processes? What common property leads to the phenomenon of Grokking?

2. How does this paper provide novel insights and practical contributions to the process of model training in real-world applications?

**Strengths And Weaknesses:**

Strengths:

The problem is interesting and can be connected with the learning theory of generalization. This paper also provides strong empirical results on synthetic data that can support the findings.

Weakness:

1. I doubt the importance of studying the Gaussian Process approach, which is simpler than neural networks. This paper appears to be a simple extension from neural networks to Gaussian processes, and I did not observe significant improvements that would warrant bringing it to the committee's attention. Utilizing observations from Gaussian processes to explain neural networks would be more interesting.

2. This paper lacks theoretical analysis.

3. The connection with the model complexity is not clearly presented.

---

> ### Author Response · Authors · 2024-01-31
> **Response**
>
> We would like to thank the reviewer for their valuable feedback.
>
> We agree with the reviewer’s points concerning the lack of theoretical analysis and the clarity of connection with model complexity. We have addressed the point regarding theoretical analysis by weakening claims regarding the scope of the paper. Instead of claiming a theory of grokking we have altered the language of our submission. Now, we only claim to suggest a mechanism for grokking which may be necessary or sufficient depending on the circumstance. To ameliorate concerns regarding the clarity of connection with model complexity, we have rewritten the introduction, streamlining the logic that we use in the paper. Additionally, if the reviewer has not had the chance, we would like to point them to Appendix B where we talk in greater detail about model complexity. The other discussions of complexity which used to be in the introduction can now be found in Appendices C and D.
>
> We do not necessarily agree, however, with the reviewer’s opinion concerning the importance of studying grokking in the Gaussian process and the simplicity of GPs. As was stated by reviewer Dkpr, grokking in GPs narrows the set of necessary conditions for grokking to occur. In addition, GPs are a cornerstone of modern machine learning and have been used for many areas such as regression, classification, latent variable modelling, black-box optimisation, active learning, and reinforcement learning. Thus, understanding grokking in this model class is crucial.
>
> Lastly, we would like to address the reviewer’s requested changes.
> 1. Regarding a high level idea why grokking occurs in both settings, we would like to point the reviewer to section 3.2 of the revised manuscript. In that section we provide a corollary and explanation as to why grokking seems to be model agnostic under our suggested grokking mechanism. Indeed, it was our discovery of grokking across multiple models which led us to develop our proposed mechanism which has the property that it is model agnostic.
> 2. The connection to real-world settings is not as close as in other machine learning research. However, we would class this work as being under the broad category of basic ML research. I.e. research that contributes towards an understanding of phenomenon in ML training dynamics which we do not understand. Nonetheless, understanding where grokking does and does not occur could be very impactful for real-world model training. What if it is the case that the same factors which lead to grokking on algorithmic datasets are also at play on some other real-world datasets? In these cases, further progress on the grokking phenomenon might lead to valuable performance gains.

---

### Review · Reviewer_Dkpr · 2023-12-02

**Summary Of Contributions:**

The authors investigate the grokking phenomenology, where networks achieve low test loss quickly with poor generalization, but continued training eventually improves generalization. They posit that the key phenomenology behind grokking is the tension between model fit and model capacity (or really, model family capacity, in some information theoretic sense). They also hypothesize that grokking is not limited to neural networks and is a more general phenomenology in machine learning.

In support of their hypothesis, the authors conduct a number of experiments. First, they construct an example of linear regression with weight norm where grokking occurs. Then they construct two GP learning scenarios where grokking occurs. They also showed that for algorithmic datasets adding spurious features ("concealment") induces grokking more easily. Finally, they use a GP example and a BNN example and study learning trajectories. The experiments provide evidence that grokking occurs when trajectories start at a place where the capacity/regularizer value is large, which leads to initial dynamics of training accuracy increasing, followed by a later time decrease in the capacity which leads to generalization.

**Audience:**

Yes

**Claims And Evidence:**

Yes

**Requested Changes:**

The main change I would like to see is further analysis of the learning dynamics of the examples in 2.1 and 2.2. I am curious whether or not these examples follow the hypotheses laid out in the rest of the paper. I think doing this analysis for the linear regression example in particular is crucial here. I am currently marking "no" in claims and evidence but am very open to changing this rating with further evidence.

Some more specific questions:

Why is the training accuracy curve in Figure 1 so discontinuous? Why, in contrast, is the test accuracy curve so smooth?

I'm not sure I understand the GP training results. My understanding is that the learning dynamics correspond to learning the parameters of the kernel, and at each step the GP inference is run with the new kernel. Is this correct? If so, why isn't the training accuracy of the GP kernel always 100%? So long as the kernel matrix is invertible the training set should be fit perfectly? Why isn't accuracy 100% at all times then?

Figure 6 is hard to parse. I can't really tell the transparency levels apart. Shouldn't there be a grokking region rather than a grokking point? Also, in the right hand side figure, it seems like the complexity drops well after learning. How does this support the hypothesis?

What do the loss plots look like in the various experiments? I am not as familiar with the grokking framework so I'm not sure how to think of the loss (training and validation) as compared to the accuracy. I believe adding this information could help readers like me who are not as familiar with the previous literature.

**Strengths And Weaknesses:**

The basic premise of the paper is sound, and the experiments are simple and focused. Demonstrating grokking in a variety of non-NN models is a valuable contribution to the literature, as it narrows down the set of necessary conditions for grokking to occur. The linking of grokking to the model capacity is also an interesting hypothesis and the experiments provide interesting evidence for this explanation for grokking.

There are two big weaknesses with the work as stands:
* In the examples where grokking occurs AND the capacity decreases at late times, much of the improvement in generalization seems to occur before the capacity drops significantly. This seems contrary to the hypothesis that the capacity decrease drives the improvement in generalization.
* It would be helpful to have more details/analysis of the learning dynamics in Sections 2.1 and 2.2. Do experimental results like those from section 2.3 also apply to 2.1 and 2.2? In particular I believe the linear regression example could do with more detailed experiments to try to show the origin of the grokking - it is a nice example due to the simplicity of the objective, and I think if findings like those in 2.3 apply to the linear regression case it would strengthen the arguments in the paper significantly.

There are also some issues with the presentation which I have flagged in "requested changes".

Edit: claims and evidence flipped to "yes" after additional experiments by the authors.

---

> ### Author Response · Authors · 2024-01-31
> **Response - Part 1**
>
> We thank reviewer Dkpr for their feedback.
>
> We would like to begin by addressing the main change requested. Namely, further analysis of the learning dynamics in the examples presented in sections 2.1 and 2.2. We agree that this kind of analysis would be a very valuable addition to the research paper and could confirm or deny the mechanism we propose in Section 3. This analysis also addresses the second weakness outlined by the reviewer. As such, we have spent time between the review and the second submission completing this analysis. Specifically, we have completed the following work:
> 1. For the linear regression case (noted as crucial in the review) we have added a new section, Appendix K, where we present three new experiments to probe the learning dynamics.
>
>      - In the first experiment, we alter weight initialisations and record three properties: the grokking gap, complexity gap and data fit gap (which we explain in the appendix). The trends in these properties are consistent with our grokking mechanism, showing clearly that model capacity plays a larger role when the grokking gap increases
>
>       - We examined the weight trajectories under various weight initialisations. We see that reversion to reduced model capacity led by weight decay seemingly produces grokking in each case.
>
>       - We removed the weight decay penalty entirely and saw that grokking failed to occur.
>
> 2. For both GP cases, we removed the complexity penalty and saw that grokking failed to occur. This can be found in the new Appendix L.
> 3. For the concealment strategy, we altered the magnitude of weight initialisation. This can be found in the new Appendix M. Under this alteration, we still saw the same trend in additional length but with a decrease in the initial grokking gap. This is consistent with our grokking mechanism (as we elucidate in that section of the appendix).
>
> We would also like to discuss the first weakness mentioned by the reviewer. Namely, that, “In the examples where grokking occurs AND the capacity decreases at late times, much of the improvement in generalisation seems to occur before the capacity drops significantly.” We would like to split our response to this weakness into the neural network and non-neural cases.
>
> In non-neural cases, our new ablation studies show that data fit alone is not sufficient to induce grokking. Instead, it seems that some capacity penalty is needed to drive grokking. As such, in these cases we can say that this observation is not contrary to the idea that capacity decrease drives the improvement in generalisation.
>
> Concerning the neural network case, we now know that grokking can occur without a complexity penalty (depending on whether one requires sharpness). As such, in this case the mechanism we suggest in Section 3 might only be sufficient or might not necessarily play a role. Nonetheless, in the BNN that we do study, it does seem in Figure 6 that at the point of grokking, the capacity changes much more significantly near the point of grokking than the data fit. The fact that it decreases further after does not discount our proposed mechanism. As has been established in other papers, in the neural network case, it seems that further capacity drop after the point of grokking results from the removal of memorising components which are no longer used to complete the problem. In the BNN the region of LELC likely corresponds to a circuit in the sense of Nanda et al. (2023) or Varma et al. (2023) rather than the whole network. As such, a relatively “small” capacity decrease can lead to grokking. More broadly, we believe that disentangling the role of regularisation in the neural network case is an important next step in the field but out of the scope of this resubmission. Indeed, our group is planning on working on this question next.

---

> ### Author Response · Authors · 2024-01-31
> **Response - Part 2**
>
> We now discuss the other questions posed by reviewer Dkpr:
> 1. Why is the training accuracy curve in Figure 1 so discontinuous? The accuracy curve in Figure 1 is discontinuous because the number of training points is very low. Alternatively, the generalisation performance is smooth as the validation dataset contains many more points.
> 2. Why isn't accuracy 100% at all times in the GP? The non-perfect accuracy of the GP classification comes from the use of the variational approximation. Specifically, both the variational posterior approximation and hyperparameters are being optimised at each step, in contrast to GP regression where we only need to deal with the hyperparameters. That is, for a given set of hyperparameters, a poor variational approximation will lead to non-perfect accuracy.
> 3. The difficulty of parsing figure 6. We have modified Figure 6 by increasing the size of the key elements of the figure (i.e. grokking points), ensuring the earlier epoch points are more transparent for clarity, and tightening our explanation of the key elements of the plot, particularly defining normalised grokking gap. We have mentioned why capacity decreases after the point of grokking is consistent with our proposed grokking mechanism in the second paragraph of this response.
> 4. Concerning the loss plots, we believe that the main loss plots are now included in the document. These include zero-one classification with linear regression, zero-one classification with GP, GP grokking on sinusoidal data, and BNN with parity prediction. Further, we have included an appendix (Appendix N) of the loss plots for grokking on all experiments with additional spurious dimensions (concealment).
>
> We believe the changes to the manuscript strengthen the evidence for the grokking mechanism we propose in Section 3. As such, we would kindly request the reviewer consider changing their assessment of “Claims and Evidence.”

---

> > ### Comment · Reviewer_Dkpr · 2024-02-16
> > **Response to authors**
> >
> > I thank the authors for their very detailed response.
> >
> > The additional experiments in Appendices K, L, and M have assuaged many of my fears about the initial set of experiments.
> >
> > In addition, I thank the the reviewers for explaining the GP classification to me. A followup question (which does not affect my review): does improving the variational approximation change the dynamics/strength of grokking?
> >
> > Figure 6 is improved. I would ask the authors to include the full loss (not just MSE) in Figure 5 - it would help give a sense of how much of the loss is due to the regularization in the various settings.
> >
> > I am switching claims and evidence to "yes" in response to the very thorough additions by the authors.

---

### Review · Reviewer_fYSj · 2024-01-04

**Summary Of Contributions:**

This paper studies the phenomenon of "grokking", which is first observed in neural network training, where the neural network achieves perfect or near-perfect accuracy on validation data after reaching similar performance on training data. In this paper, the authors empirically present the grokking phenomenon in other models like Gaussian process (GP) classification and regression, linear regression, and BNNs. The authors also propose that grokking can be influenced by adding dimensions containing spurious information to datasets and suggest it's not specific to certain types of optimization like stochastic gradient descent (SGD) or weight norm regularization. Following these observations, the authors introduce the complexity theory of grokking and claim that the phenomenon of grokking should be model agnostic.

**Audience:**

Yes

**Claims And Evidence:**

Yes

**Requested Changes:**

1. There are some other papers theoretically analyzing the grokking phenomenon, e.g., [1-3]. Some references should be updated and you need to have some discussions related to these references.
2. [2] has already theoretically studied the grokking phenomenon in linear networks which could be related to your linear regression result.
3. When you use Dataset 1-10 in the main text, you should mention that the detail is referred to Appendix C.
4. What is $w^i$ in (8)? You need more explanation.
5. The structure of Section 1 appears somewhat disorganized and could benefit from clearer organization to enhance the readability and flow of the information presented. You may need to summarize the main contribution of this paper for clarity in this section as well.

[1] Grokking as the Transition from Lazy to Rich Training Dynamics

[2] Grokking in Linear Estimators--A Solvable Model that Groks without Understanding

[3] Dichotomy of Early and Late Phase Implicit Biases Can Provably Induce Grokking

**Strengths And Weaknesses:**

The paper presents new empirical evidence for the phenomenon of grokking and extends the understanding of this phenomenon to non-neural architectures. These results can be a complement to previous results focusing on neural network models. It introduces a new data augmentation technique that binds grokking, which could be a significant contribution to the field.

However, this paper lacks theoretical understanding and explanations of the grokking phenomenon. Introducing experiments in GP and BNNs currently does not help us fully understand the emergence of grokking in the training process. The authors claim they have an effective theory of grokking but there is only empirical observation and no mathematical theory proved in the paper, especially in Section 3.

---

> ### Author Response · Authors · 2024-01-31
> **Response**
>
> We thank reviewer fYSj for their feedback.
>
> We would like to begin by commenting on the broad strengths and weaknesses identified. We agree generally with the assessment of the paper. Namely, that useful empirical results have been uncovered but that we have not provided a mathematical framework for the grokking phenomenon. It seems clear now that our claims in the original manuscript were too strong. To combat this we have made two changes:
> 1. We have softened the language of the paper. Where before section 3 described a theory of grokking it now discusses a possible mechanism for grokking.
> 2. We have provided further ablations and analysis which supports the mechanism that is described in section 3 (see new text in Section 2 and Appendices K, L and M)
>
> We now wish to comment on how we have implemented the requested changes (1)-(5):
> 1. We have included references [1]-[3]. In particular, we explain these papers in the introduction and discuss their relevance in a new paragraph within the future work section. The reason we did not include them earlier was their concurrent or subsequent release to this submission. Indeed, references [2] and [3] were published on arxiv after we submitted this paper to TMLR.
> 2. While reference [2] studies grokking in linear models, it does so by taking the accuracy measure to be the error function whose input is a threshold divided by the training or validation mean squared error between linear teachers and students. This is quite distinct from our case where we use weight decay and sign-based accuracy. Indeed, as is presented in Appendix K, we require weight decay for grokking in our setting whereas [2] does not. As such, it seems that a different process is driving grokking in their case when compared to ours.
> 3. We have ensured any reference to Datasets 1-10 also includes an explicit and physical link to Appendix C.
> 4. We have expanded our explanation of Equation (8) (now Equation 3) so that we fully elicit what each individual term means, particularly with respect to $w^i$
> 5. We have made several changes in section 1 to improve readability and flow. Specifically, we have:
>     * Moved the discussion of all complexity measures to the appendix.
>     * Removed the separation of definitions and paragraphs, instead explaining things inline.
>     * Expanded discussions of current theories of grokking.
>     * Provided a clear set of contributions we have made which make our work distinct from previous investigations of grokking.

---

### Author Response · Authors · 2024-01-31
**General comment**

Having received a valuable set of reviews, we have updated our manuscript accordingly. We have moderated our claims about the scope of our proposed grokking mechanism, no longer describing it as a theory. Further, we have added substantial empirical evidence to support the mechanism that we propose. This evidence includes: ablative experiments where we fail to see grokking without a complexity measure, a substantial analysis of the linear regression setting with three new experiments, and a further analysis of our proposed data augmentation technique under different initialisations (see Appendices K, L, and M). We've also updated the manuscript to incorporate new references, specifically those published concurrently or subsequent to our initial submission, and have modified the introduction to better contextualise our work within the existing literature. Finally, we have made some cosmetic changes, including improving the clarity of figures, expanding on unclear explanations, and improving the overall structure of the paper. These revisions aim to directly address the reviewers' critiques, clarifying our contributions to the understanding of grokking and its implications for machine learning research.

---

### Decision · Action_Editor_Z8Sh · 2024-02-21

**Recommendation:** Accept with minor revision

**Comment:**

See above

**Audience:**

To help transparent decision-making, I include here the TMLR's criteria:

- *Are the claims made in the submission supported by accurate, convincing, and clear evidence?*
- *Would at least some individuals in TMLR's audience be interested in knowing the findings of this paper?*

As stated in the "Claims and Evidence" section, while one reviewer found the topic super interesting, another reviewer raised a concern whether ICLR's audience would find this work engaging, in terms of real-world application. While this comment is about ICLR's audience, it has value to be considered by the authors if they want to slightly edit their paper towards this direction (e.g., by adding discussions or highlighting existing ones on how this work affects practical scenarios found in real cases).

**Claims And Evidence:**

To help transparent decision-making, I include here the TMLR's criteria:

- *Are the claims made in the submission supported by accurate, convincing, and clear evidence?*
- *Would at least some individuals in TMLR's audience be interested in knowing the findings of this paper?*

Based on the updated empirical results and the discussions among the reviewers, the paper considers an interesting topic that is well-supported by empirical evidence (as promised). While the theoretical part of grokking is missing, the paper is interesting and can be connected with the learning theory of generalization. This paper also provides strong empirical results on synthetic data that can support the findings.

Based on the final recommendations by the reviewers, it is clear that the paper passes the bar of acceptance.

---

> ### Author Response · Authors · 2024-03-19
> **Added Camera-Ready Version**
>
> We have added the camera-ready version. The manuscript now includes a short subsection on real world applications, an acknowledgement section and a supplementary materials section which links to the relevant GitHub repository.
>
> In addition, we have produced a video for the paper which is now linked in the submission.
>
> We would again like to thank the action editor and reviewers for their help in improving our paper.